# DISENTANGLING LOCALITY AND ENTROPY IN RANKING DISTILLATION

## ABSTRACT

The training process of ranking models involves two key data selection decisions: a sampling strategy (which selects the data to train on), and a labeling strategy (which provides the supervision signal over the sampled data). Modern ranking systems, especially those for performing semantic search, typically use a "hard negative" sampling strategy to identify challenging items using heuristics and a distillation labeling strategy to transfer ranking "knowledge" from a more capable model. In practice, these approaches have grown increasingly expensive and complex—for instance, popular pretrained rankers from SentenceTransformers involve 12 models in an ensemble with data provenance hampering reproducibility. Despite their complexity, modern sampling and labeling strategies have not been fully ablated, leaving the underlying source of effectiveness gains unclear. Thus, to better understand why models improve and potentially reduce the expense of training effective models, we conduct a broad ablation of sampling and distillation processes in neural ranking. We frame and theoretically derive the orthogonal nature of model geometry affected by example selection and the effect of teacher ranking entropy on ranking model optimization, establishing conditions in which data augmentation can effectively improve bias in a ranking model. Empirically, our investigation on established benchmarks and common architectures shows that sampling processes that were once highly effective in contrastive objectives may be spurious or harmful under distillation. We further investigate how data augmentation—in terms of inputs and targets—can affect effectiveness and the intrinsic behavior of models in ranking. Through this work, we aim to encourage more computationally efficient approaches that reduce focus on contrastive pairs and instead directly understand training dynamics under *rankings*, which better represent real-world settings.

## 1 INTRODUCTION

Pre-trained language Models (PLMs) (Vaswani et al., 2017; Devlin et al., 2019) have been shown to be effective in ad-hoc ranking tasks (Lin et al., 2021). By training on large labeled datasets (Nguyen et al., 2016), they can often outperform classical term-weighting models (Nogueira & Cho, 2019). Since these early works, a key direction in improving PLM-based ranking models has been improving their data augmentation pipelines, which typically now combine "hard" negative mining (Karpukhin et al., 2020; Qu et al., 2021) (the deliberate selection of challenging non-relevant texts), and distillation of relevance estimation from an existing teacher model (Hinton et al., 2015; Lin et al., 2020; Hofstätter et al., 2020). While these techniques have independently demonstrated clear gains in representation learning (Hsu et al., 2021), their interaction in ranking distillation settings where there is no explicit notion of a negative is poorly understood and is applied in several works with little or no ablation (Xiao et al., 2022; Ren et al., 2021; Song et al., 2023; Wang et al., 2024).

In Information Retrieval (IR), hard negatives are typically sampled from candidate sets scored by one or more initial models (Karpukhin et al., 2020; Gao et al., 2021). In contrastive objectives, increasing the locality of (reducing the distance between) the candidate set creates a more challenging classification task (Gutmann & Hyvärinen, 2010; Ceylan & Gutmann, 2018) as illustrated in Figure 1, often improving downstream effectiveness. Distillation, in turn, replaces binary labels with soft targets from a teacher (Bucila et al., 2006; Ba & Caruana, 2014; Hinton et al., 2015), allowing students to match or surpass larger estimators (Pradeep et al., 2022; Xiao et al.,

2022; Pradeep et al., 2023). Contemporary systems couple the two: teachers are employed both to score and to label documents, producing multistage cascades of models that must be trained and queried at scale to collect training data. A common example is the five-stage *SentenceTransformers* pipeline whose twelve cross-trained models are further filtered by a classifier (Reimers & Gurevych, 2019), sometimes upon previous iterations of each other. Such complexity is problematic. Each additional model inflates computational cost and $CO_2$ footprint (Scells et al., 2022) and hinders reproducibility (Wang et al., 2022). From a theoretical standpoint, negative-selection heuristics influence only which instances are labelled, not the Lipschitz-geometry or teacher-entropy terms that influence generalization (cf. 2.1), thus the case illustrated in Figure 1 often does not occur.

Furthermore, existing work on increasing the "hardness" of negatives often focuses solely on the domain of a ranking (Karpukhin et al., 2020; Gao et al., 2021; Qu et al., 2021), either using expensive ensembling approaches or iteratively choosing challenging domains based on heuristics such as model uncertainty (Xiong et al., 2021; Zhan et al., 2021). Thus, much focus is on reducing epistemic uncertainty, which may lead to a highly confident model while neglecting the irreducible aleatoric uncertainty inherent in relevance labels. Such is the appeal of distillation that multiple relevant documents may be present and optimised within a single instance, as illustrated in Figure 1.

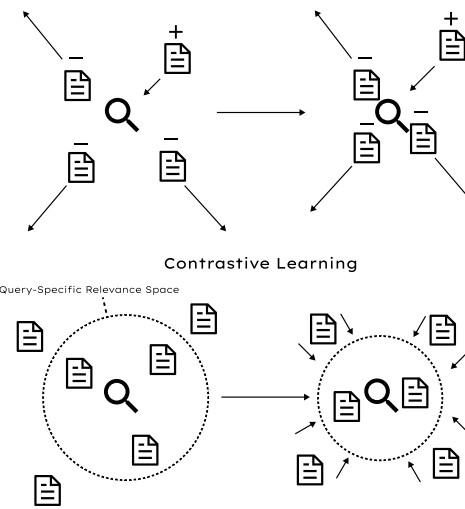

Figure 1: Illustration of the requirement of different learning paradigms to improve effectiveness. Crucially, if the choice of $\mathcal{X}$ (including negatives) does not tighten the metric-measure space over the given query, effectiveness will not improve.

The notion of negative mining in ranking tasks is largely an artifact of contrastive objectives, often applied in representation learning. This nomenclature can be extended to functions such as ad-hoc search and reinforcement learning from human feedback (Christiano et al., 2017; Rafailov et al., 2023); however, the increasing use of distillation or explicit annotation means that we operate over explicit rankings instead of solely positive-negative pairs. Thus, both theoretically and empirically, we investigate common training settings controlling for heuristics governing example locality and the entropy of target distributions. We underpin empirical ablations with a generalisation bound over ranking distillation (Sec. 3) whose bias term depends exclusively on (i) the intrinsic diameter of the query manifold and (ii) the teacher's pairwise entropy—two quantities unchanged by additional negative-mining stages.

In isolating the contributions of ranking domains and target distributions in training settings, our primary contributions are two-fold: *1) We show complex pipelines to be largely identical in effectiveness to naive approaches under semi-supervision. 2) We provide clear empirical evidence of the causal factors in model effectiveness when applying data augmentation.*

In investigating these factors, we aim to focus research on factors of effectiveness that are appropriate given a *particular* search setting and reduce the spurious training and inference of multiple models.

## 1.1 RELATED WORK

**Data Selection in Ranking** Neural ranking models based on PLMs often apply contrastive objectives which benefit from the selection of similar instances, as these objectives become more challenging when examples are more similar in a geometric space but are different in terms of the target objective. Gao et al. (2021) argued that greater locality and a greater number of samples inspired by NCE (Gutmann & Hyvärinen, 2010) could further enhance the benefit of localized negatives, finding that negatives sampled from more precise rankings yielded greater effectiveness, which scales with the number of samples. Additionally, several approaches have been proposed to apply active learning to select negative examples which are most challenging to the model, which, while being quite effective, require a sufficient number of annotated queries to yield significant results (Althammer

et al., 2023). Several works propose that the flaw in negative mining is that as we produce an ever stronger source of negatives, due to label sparsity, we inevitably sample false negatives; as such, the notion of de-noised negatives has been proposed (Qu et al., 2021) and has become increasingly complex (Reimers & Gurevych, 2019).

Distillation in some form has been applied both as weak supervision Dehghani et al. (2017) and more commonly from a larger model in ranking (Lin et al., 2020; Hofstätter et al., 2020; Qu et al., 2021; Xiao et al., 2022). For distillation minimal investigation has occurred to our knowledge to select an appropriate domain. Initial approaches adopted similar negative mining strategies those in contrastive settings (Lin et al., 2020; Hofstätter et al., 2020), increasingly selection moves towards top-$k$ elements (Pradeep et al., 2023; Schlatt et al., 2025) as opposed to sampling due to both computational expense of large teacher models (Sun et al., 2023a) where a larger number of elements would not be ranked without being utilised and due to the ever increasing precision of PLM-based approaches (Sun et al., 2023b). For parity with contrastive objectives and analysis, we investigate the sampling of elements instead of top-$k$.

**Understanding Distillation** Knowledge distillation aims to transfer task effectiveness from one model to another. In ranking this approach has been applied extensively both as a weak-supervision signal for bootstrapping early attempts at PLM-based ranking and for efficiently training lightweight ranking models. Though some investigations provided qualitative explanations for the effectiveness of knowledge distillation (Ba & Caruana, 2014; Hinton et al., 2015), recent work has focused on more principled justifications of distillation and divergence-based learning objectives. The effectiveness of a teacher-student distillation setting have been examined in terms of intrinsic task difficulty (Ji & Zhu, 2020) and the degree to which a student follows a teacher distribution (Nagarajan & Kolter, 2019) suggesting that divergence from a teacher distribution is not a problem in model generalisation as shown empirically by methods in retrieval which ignore the exact densities of discrete ranking distributions (Pradeep et al., 2023; Schlatt et al., 2025). In terms of explicit generalisation bounds, Hsu et al. (2021) provide a bound under uniform convergence, using distillation as a vector for understanding the original teacher model, we diverge from this setting as in downstream Information Retrieval we focus on trading off effectiveness for reduced latency. In terms of representation learning, the theoretical implications of data selection by an existing model have been explored (Lin et al., 2024) however this work does not extend to divergence-based losses explored in this work.

## 2 THEORETICAL ANALYSIS

We now formalise the contribution of data augmentation to ranking model optimisation. We provide a generalisation bound in terms of sample locality and target entropy towards understanding where data selection can improve effectiveness.[1]

### 2.1 PRELIMINARIES

**The Ranking Problem** Given a corpus of texts, $\mathcal{C} = \{D_i\}_1^{|\mathcal{C}|}$ and a query $Q$, a top-$k$ ranking $\mathcal{R} = [D_i]_1^k$ (where $k \ll |\mathcal{C}|$) ordered by estimated relevance to $q$ determined by estimates some model $f : \mathcal{X} \to \mathbb{R}$ where $\mathcal{X} \equiv (Q, D)$. A learned ranking model is often modeled as an unnormalized estimator of $p(D|Q)$ (Robertson et al., 1995),[2] modeling the likelihood of $D$ being relevant to $Q$. We focus on two common architectures, cross-encoders (Nogueira et al., 2020) and bi-encoders (Karpukhin et al., 2020). A cross-encoder treats ranking as a regression over the joint representation of a query and document encoded by a PLM from which a relevance score is estimated. A bi-encoder instead separately encodes queries and documents, treating ranking as a maximum inner-product search problem over pooled latent representations.

**Training Ranking Models** Let $\mathcal{Q}$ denote a set of training queries. For each $Q \in \mathcal{Q}$ we observe a finite candidate list $\mathcal{D}_Q$ commonly treated as pairs $\mathcal{X}_Q = \{(Q, D_i) : \forall D_i \in \mathcal{D}_Q\}$. Each element of $\mathcal{X}_Q$ can be assigned a binary relevance label $\mathcal{Y}_Q = \{y_i\}_1^{|\mathcal{X}_Q|}, y \in \{0, 1\}$. Commonly, solely 'positive' candidates are explicitly annotated (i.e., $y \leftarrow 1$) (Nguyen et al., 2016), forming a labelled set $\mathcal{L}$. All other elements are sampled from a larger set of similar documents chosen by a heuristic, forming

---

[1] A full notation table can be found in Appendix A

[2] $D$ may be one or more texts depending on architecture

a pseudo-negative set $\mathcal{U}$. In distillation settings, targets $\mathcal{Y}_Q$ are instead determined by an existing teacher model $g$ such that $\mathcal{Y}_Q = \{g(Q, D), \forall (Q, D) \in \mathcal{X}_Q\}$.

**Choosing Pairs** It is common to condition the sampling of pseudo-negative examples on some existing scorer (Karpukhin et al., 2020; Qu et al., 2021). Formally, let $\mu_Q$ define a query-specific measure over $\mathcal{X}$, and let $\nu_Q$ denote a biased measure derived from some existing model for data selection. Recent approaches use an ensemble of systems (Qu et al., 2021; Song et al., 2023) inducing $\nu_Q$, which, in contrastive settings, can be further filtered by a model $g$ to ensure that $d(\mathcal{X}_i, \mathcal{X}_j) > \epsilon, \forall \mathcal{X}_{j, i \neq j} \in \mathcal{X}$. This assumes that $\mathcal{X}_i$ can be a labeled positive for the contrastive objective. This step, given its expense and implied confidence in the discriminative ability of $g$, would imply that one should learn an approximation of the teacher model $g$ in a semi-supervised setting as opposed to solely filtering a candidate distribution. Nevertheless, this approach of ensembling and filtering is frequently employed, and thus we consider it for completeness.

**Problem Setting** The subjectivity of relevance and the scale of modern web-scale corpora make estimation of a ranking *intrinsically subjective* (Voorhees, 1998; Parry et al., 2025): for most queries, we neither observe a complete ordering of candidates nor can we assume perfect recall in finding relevant documents, thus there can be many relevant documents beyond the single-relevant document constraint of contrastive learning. We therefore often work with *a)* a sparse set of human judgements and noisy negative examples, or *b)* a teacher model $g : \mathcal{X} \to \mathbb{R}$ trained on auxiliary data. The objective of student $f : \mathcal{X} \to \mathbb{R}$ is to rank elements from a structured space $\mathcal{X}$ equipped with metric $d$. Given a query $Q$, the goal is to learn a scoring function that produces rankings aligned with a teacher model $g$.

Recall that for each query $Q \in \mathcal{Q}$ we have a candidate pairs $\mathcal{X}_Q = \{(Q, D_1), \ldots, (Q, D_{m_Q})\}$ together with labels $\mathcal{Y}_Q \in \{y_0, y_1, \ldots, y_{m_Q}\} \cup \{\varnothing\}$. A student ranker $f : \mathcal{X} \to \mathbb{R}$ outputs real-valued scores whose descending order defines the predicted ranking. Our loss criteria will contrast query-document pairs such that we look to minimise the pair-wise risk of misordering two pairs compared to their order under model $g$. Metric structure matters fundamentally because our student learns in a setting where available training data represents a limited and often biased sample from the true space of relevant elements around an anchor query. The geometry constrains how knowledge can transfer between observed pairs.

Thus, our problem lies in the contribution of data augmentation to the effectiveness of ranking models. Distillation through criteria such as RankNet (Burges, 2010) blurs the boundary between classical notions of contrastive learning and knowledge distillation. No score values are used, and technically, contrastive objectives do the same, albeit with arbitrary negative ordering, both conditioned on some teacher (or negative miner) $g$. Negative sampling and knowledge distillation can be seen as orthogonal; sampling provides suitable observations given a downstream task and requires some labeling process for optimization; distillation provides the labeling process for optimization, but does not provide an explicit selection process for observations.

### 2.2 NOTATION AND DEFINITIONS

Assuming student model $f$ and teacher model $g$, we use the output of model $g$ as targets $\mathcal{Y}$. We define locality in terms of a query-specific measure $\mu_Q$ over our input space $\mathcal{X}$ modelling the geometry of relevant elements conditioned on $q$. Formally, let $(\mathcal{X}, \mu_Q, d)$ be a metric–measure space with complete space $\mathcal{X}$, measure $\mu_Q$ conditional on $Q$ and distance $d$ (we use cosine distance over latent representations), define the essential diameter of this space as

$$\Delta_Q = \operatorname*{ess\,sup}_{(x,x') \in \mathcal{X}^2} d(x, x') \text{ with respect to } \mu_Q \otimes \mu_Q. \tag{1}$$

Hypothesising that higher-entropy ranking targets encode additional useful information, similarly to the propositions of Hinton et al. (2015), we model the entropy of a teacher ranking under pair-wise preferences. We investigate losses that can be seen as Bregman divergences as they are prevalent in the empirical ranking literature (RankNet (Burges, 2010), MarginMSE (Hofstätter et al., 2020), KL-divergence (Kullback & Leibler, 1951)) and admit clean theoretical analysis. Define the pair-wise risk of $f$ as:

**Definition 2.1** (Pair-wise Risk). *For a scorer $f$ and query measure $\mu_Q$, the* pair-wise risk *is the probability of mis-ordering:*

$$\mathcal{R}_{\mu_Q}(f) = \Pr_{(x,x') \sim \mu_Q^{\otimes 2}}[f(x) < f(x')] = \mathbb{E}_{\mu_Q^{\otimes 2}}\big[\mathbf{1}\{f(x) < f(x')\}\big]. \tag{2}$$

To minimise this risk, we use a distillation loss in the form of a Bregman divergence.

**Definition 2.2** (Bregman Distillation Loss). *For convex potential $\phi$, student $f$ and teacher $g$, the distillation loss on pair $(x, x')$ is*

$$\ell(f, g; x, x') = D_\phi(f(x) - f(x') \| g(x) - g(x')), \tag{3}$$

*where $D_\phi(a\|b) = \phi(a) - \phi(b) - \phi'(b)(a - b)$ is the Bregman divergence.*

Under this pair-wise setting, we define:

**Definition 2.3** (Query Entropy). *For teacher $g$ and query measure $\mu_Q$, let $p_{x,x'} = \Pr[g(x) > g(x')]$. The* query entropy *is*

$$H(g) = \mathbb{E}_{(x,x')\sim\mu_Q^{\otimes 2}}[-p_{x,x'} \log p_{x,x'} - (1 - p_{x,x'}) \log(1 - p_{x,x'})]. \tag{4}$$

When entropy is too low, this can be seen as a trivial setting under a Bregman divergence criterion, empirically leading to collapse as the optimisation task is too easy. As elements become more difficult to distinguish as entropy increases, we seek to understand how the domain over which a ranking is calculated affects optimisation in settings where entropy is high. To measure the contribution of this entropy to the optimisation of a student we follow Painsky & Wornell (2020), we use the misordering probability of model $g$ via Pinsker's inequality:

**Definition 2.4** (Misordering probability under entropy (Proof in Appendix B)). *Following Painsky & Wornell (2020), For teacher $g$ with pair-wise entropy $H(g)$, the misordering probability satisfies*

$$\eta(H(g)) = \frac{1}{2} - \sqrt{\frac{\log 2 - H(g)}{2}}. \tag{5}$$

## 2.3 GENERALISATION UNDER DATA AUGMENTATION

We look to understand the contribution of these data selection factors to model optimisation in tandem. Thus, we establish a generalization bound for the special case of ranking distillation through the excess risk of learning from a teacher under a particular sampling policy. We apply a PAC bound; these bounds affect a model's preference for a particular hypothesis within a hypothesis class, effectively governing how a model will generalise, with our key novelty being the derivation of a bias term conditioned on data augmentation factors.

**Theorem 2.1** (Ranking Distillation Generalisation Bound (Proof in Appendix C)). *Let $(\mathcal{X}, d, \mu_Q)$ be a metric-measure space for $Q$ and let $\mathcal{H}$ be a hypothesis class of VC dimension $d$ such that every $h \in \mathcal{H}$ is $L$-Lipschitz. Let $f^\star = \arg\min_{h\in\mathcal{H}} \mathcal{R}_{\mu_Q}(h)$ and let $\widehat{f}$ minimise an empirical Bregman loss with convex potential $\phi$. Then for every confidence level $\delta \in (0, 1)$, with probability at least $1 - \delta$,*

$$\mathcal{R}_{\mu_Q}(\widehat{f}) - \mathcal{R}_{\mu_Q}(f^\star) \leq \zeta L \Delta_Q \eta(H(g)) + C\sqrt{\frac{d \log(1/\delta)}{n}}, \tag{6}$$

*where $\zeta$ depends only on divergence potential $\phi$, $C > 0$ is an absolute constant, and $n$ is the number of observed pairs.*

**Implications.** This bound indicates that in order to optimise a tighter query-specific measure space (akin to "harder" negatives), an increasingly confident (and accurate) teacher is required. Additionally, if "harder" negatives do not tighten the diameter $\Delta_Q$, they will not yield greater generalization. Incorporating unlabeled data through distillation can improve performance by yielding more accurate estimates of the class diameter, and although the choice of Bregman loss affects the constant $\zeta$ in the bound, it does not alter the fundamental scaling behavior. The metric structure manifests through the essential diameter $\Delta_Q$, which captures how spread out the relevant items are around each query. This geometric constraint affects generalization in biased sampling scenarios such as negative mining. Consequently, a biased sampler, where pairs are not drawn under the true measure $\mu_Q$, should be explicitly accounted for.

## 2.4 Effect of Biased Sampling: Density-Ratio Adjustment

In practice, we often use biased sampling strategies to select training examples. Let $\pi_Q$ denote the mining/retrieval policy with $\mathrm{supp}\,\pi_Q \subseteq \mathrm{supp}\,\mu_Q$. Following Hsu et al. (2021), define the density ratio bridging to $\mu_Q$ by

$$\kappa_Q = \underset{x \in \mathrm{supp}\,\pi_Q}{\mathrm{ess\,sup}} \frac{d\mu_Q}{d\pi_Q}(x) < \infty. \tag{7}$$

**Corollary 2.1** (Fixed-miner density–ratio bound (Proof in Appendix D)). *Under a biased sampling policy with density ratio $\kappa_Q$ relative to $\mu_Q$, the excess risk bound in Theorem 2.1 becomes*

$$\mathcal{R}_{\mu_Q}(\widehat{f}) - \mathcal{R}_{\mu_Q}(f^\star) \;\le\; \zeta\, L\Delta_Q\, \eta\big(H(g)\big) \;+\; C\,\kappa_Q\,\sqrt{\frac{d\,\log(1/\delta)}{n}}. \tag{8}$$

When the pool is *in-domain* ($\kappa_Q \approx 1$), the rate matches the unbiased case; off-domain pools inflate the bound linearly, adding risk as a chosen sampling policy diverges from the ideal policy. Empirical values of $H(g)$, $\Delta_Q$, and $\kappa_Q$ are reported in Table 2.

## 3 Empirical Investigation

Having established theoretical factors in ranking distillation, we now investigate common settings from literature.

### 3.1 Experimental Setup

**Datasets and Metrics** We assess in-domain effectiveness on the TREC Deep Learning 2019 (Craswell et al., 2020b) and 2020 (Craswell et al., 2020a) test collections, retrieving from the MSMARCO passage collection (Nguyen et al., 2016). To assess out-of-domain effectiveness, we use all public test collections comprising the BEIR benchmark (Thakur et al., 2021). We report dataset statistics and full out-of-domain results in Appendix F. All reported test collections apply normalized discounted cumulative gain (nDCG) (Järvelin & Kekäläinen, 2002) as their primary measure and we report this value at a rank cutoff of 10 as is standard, to provide insight at greater recall values, we also report mean averaged precision (MAP) over full rankings. We use a TOST (Two One-Sided T-test) to assess statistical equivalence with $\alpha = 0.05$ and means bounded at $1\%$. Further details are provided in Appendix E.

**Models** We train both cross-encoders and bi-encoders to assess how representation capacity affects investigated training settings. All cross-encoders are initialised from an ELECTRA checkpoint (Clark et al., 2020), as this model has been shown empirically to yield greater effectiveness than a standard BERT model for cross-encoder initialisation (Pradeep et al., 2022). We initialise all bi-encoders from the original BERT checkpoint (Devlin et al., 2019). Each model is a standard transformer encoder with twelve layers and six attention heads per layer. To ensure reproducibility and clear attribution of effectiveness, we train a cross-encoder following (Pradeep et al., 2022) using the ELECTRA architecture trained for one epoch with BM25 localised negatives on the MSMARCO-passage training set. Under distillation, it is generally unnecessary (Althammer et al., 2023; Schlatt et al., 2025) and computationally infeasible in our study to train all model variants at this scale; thus, this model acts as a teacher trained in a data-rich environment.

**Loss Criteria** As a supervised criterion, we employ localized contrastive estimation (LCE) (Gao et al., 2021), similar to infoNCE (Gutmann & Hyvärinen, 2010) and more generally model-conditioned NCE (Ceylan & Gutmann, 2018). This criterion, assuming $\mathcal{X}_i$ is a known positive, is defined as:

$$\ell_{\mathrm{LCE}}(\mathcal{X};\, f) = -\log \frac{\exp\big(f(\mathcal{X}_i)/\tau\big)}{\exp\big(\mathcal{X}_i/\tau\big) + \sum_{j=1, j\neq i}^{m-1} \exp\big(\mathcal{X}_j/\tau\big)}. \tag{9}$$

We employ three semi-supervised criteria prevalent in neural ranking literature. The first marginMSE aims to reduce the effect of differences in ranking modes by optimising the margin between positive ($\mathcal{X}_i$) and negative ($\mathcal{X}_j$) elements instead of pointwise scores (Hofstätter et al., 2020).

$$\ell_{\mathrm{marginMSE}}(\mathcal{X}; f, g) \;=\; \sum_{\substack{j=1 \\ j\neq i}}^{m} \Big[ f(\mathcal{X}_i) - f(\mathcal{X}_j)) - g(\mathcal{X}_i) + g(\mathcal{X}_j) \Big]^2 \tag{10}$$

The second, RankNet, is common in learning-to-rank literature (Burges, 2010). It sees increasing application with increasing precision of modern ranking models as it optimizes all $x, x'$ interactions agnostic of human labels.

$$\ell_{\text{RankNet}}(f) = \sum_{i=1}^{|\mathcal{X}|} \sum_{\substack{j=1 \\ j \neq i}}^{|\mathcal{X}|} -\Big[ y_{ij} \, \log \sigma(s_{ij}) + (1 - y_{ij}) \log\big(1 - \sigma(s_{ij})\big) \Big], \tag{11}$$

where $s_{ij} = f(\mathcal{X}_i) - f(\mathcal{X}_j)$, $\sigma(z) = \frac{1}{1+e^{-z}}$, and $y_{ij} \in \{0, 1\}$.

Finally, KL divergence is commonly used as a loss criterion in several settings beyond ranking (Lin et al., 2020; Kingma & Welling, 2014). It assumes, much like RankNet, that a reference distribution, in this case $[g(\mathcal{X}_i)]_1^m$, represents the absolute ground truth.

$$\ell_{\text{KL}}(\mathcal{X}; f, g) \;=\; \sum_{j=1}^m f(\mathcal{X})_j \, \log \frac{f(\mathcal{X})_j}{g(\mathcal{X})_j} \tag{12}$$

We show these semi-supervised criteria can be expressed as Bregman divergences in Appendix A.

**Implementation Details** All models observe a total of $12M$ documents from the MSMARCO-passage collection(Nguyen et al., 2016), providing approximately equal computational budget across all training settings with a fixed batch size of 128 pairs. We follow (Pradeep et al., 2022) in setting the learning rate of cross-encoder runs to $1e{-}5$ and (Hofstätter et al., 2020) for bi-encoders, setting the learning rate to $7e{-}6$. We apply a learning rate warmup for 0.1 epochs and then linear decay with an AdamW optimizer using default hyperparameter settings (Loshchilov & Hutter, 2019). We use the Transformers library (Wolf et al., 2020) with a PyTorch backend (Paszke et al., 2019) for all training processes. We use PyTerrier for inference and evaluation (Macdonald & Tonellotto, 2020).

**Negative Sampling Sources** We consider four sampling sources in our investigation, aligning with those applied in literature. The first is uniform selection from the training corpus (Random). The second is a lexical heuristic BM25 ($k_1 = 1.2$, $b = 0.75$) (Robertson et al., 1995), a lightweight retrieval model. The third is our teacher model, a cross-encoder (CE). Finally, we apply the ensemble pipeline shown in Figure 1 (Ensemble). We use the rankings supplied, but filter them by our teacher model to ensure fairness across settings. We outline all models contained within this ensemble in Appendix E.

### 3.2 DISCUSSION

**Effectiveness under Different Empirical Distributions** Our ablation of sampling distributions under semi-supervision shows multiple cases where investing computational budget in a strong estimator is unnecessary to improve generalisation both in- and out-of-domain. In Table 1, rows 1-4 show effectiveness in a supervised setting using a contrastive objective. In-domain, it is clear that localised sampling by heuristics can be effective as there is a clear trend in effectiveness as "hardness" and thus computation is expended. Out-of-domain, observe that generally, where a single estimator-induced distribution is insufficient to cover the query manifold, a random or ensemble sample yields greater robustness. We find that, though in explicit contrastive learning, there is a clear trend that the tighter a sampling distribution is, the more model effectiveness can continue to improve; we find that under all semi-supervision settings, effectiveness plateaus once a minimal locality is enforced (BM25) with inconsistent effectiveness improvements suggesting that heuristics such as mining from rankings are insufficient to explain effectiveness gains. Though bias induced by sampling is indeed reduced by ensembling approaches, empirical values of the query-specific diameter show that the query space does not become more compact. This is shown in Table 2, explaining minimal change in out-of-domain effectiveness as our bias term is largely unchanged across domains. However, aligning with corollary 3.1.1, we see that when density ratios are minimised across our settings under an ensembling approach, a Bi-Encoder continues to improve, suggesting this term can have a larger effect; nevertheless, we do not find settings where a statistically differentiable positive effect is found across multiple domains (e.g helping both in and out-of-domain) when applying computationally expense data selection strategies.

**Effectiveness under Different Entropy** Having controlled for entropy in different training settings, we now fix the sampling domain $\nu_Q$, choosing BM25 and select rankings based on where they lie

Table 1: Ranking effectiveness across loss criteria and sampling domains. In-domain effectiveness is evaluated on TREC Deep Learning test collections, and out-of-domain effectiveness is evaluated on the BEIR benchmark (per-dataset effectiveness is shown in Appendix F). Superscripts denote statistical equivalence via TOST (1% bound, $\alpha = 0.05$).

| | | TREC DL'19 | | | | TREC DL'20 | | | | BEIR | |
| | | BE | | CE | | BE | | CE | | BE | CE |
| Loss | Domain | nDCG | MAP | nDCG | MAP | nDCG | MAP | nDCG | MAP | nDCG | nDCG |
| LCE | Random | 0.546 | 0.333 | 0.628[bcd] | 0.407[bcd] | 0.523 | 0.360 | 0.615 | 0.406 | 0.459[bc] | 0.507[d] |
| LCE | BM25 | 0.642[cd] | 0.385[cd] | 0.681[acd] | 0.459[acd] | 0.622[cd] | 0.425[cd] | 0.686[cd] | 0.488[cd] | 0.462[ac] | 0.472 |
| LCE | CE | 0.634[bd] | 0.389[bd] | 0.675[abd] | 0.454[abd] | 0.632[bd] | 0.424[bd] | 0.689[bd] | 0.482[bd] | 0.456[ab] | 0.478 |
| LCE | Ensemble | 0.658[bc] | 0.397[bc] | 0.730[abc] | 0.501[abc] | 0.655[bc] | 0.448[bc] | 0.738[bc] | 0.520[bc] | 0.459 | 0.502[a] |
| RankNet | Random | 0.336 | 0.199 | 0.616 | 0.409 | 0.273 | 0.185 | 0.578 | 0.404 | 0.323 | 0.445 |
| RankNet | BM25 | 0.653[cd] | 0.410[cd] | 0.731[cd] | 0.485[cd] | 0.659[cd] | 0.424[cd] | 0.739[cd] | 0.498[cd] | 0.468 | 0.527[c] |
| RankNet | CE | 0.657[bd] | 0.409[bd] | 0.721[bd] | 0.489[bd] | 0.651[bd] | 0.432[bd] | 0.734[bd] | 0.496[bd] | 0.475 | 0.526[b] |
| RankNet | Ensemble | 0.679[bc] | 0.413[bc] | 0.719[bc] | 0.483[bc] | 0.689[bc] | 0.459[bc] | 0.747[bc] | 0.508[bc] | 0.494 | 0.488 |
| mMSE | Random | 0.602[bcd] | 0.374[bcd] | 0.693[bcd] | 0.433[bcd] | 0.637[bcd] | 0.415[bcd] | 0.685[bcd] | 0.470[bcd] | 0.459 | 0.477 |
| mMSE | BM25 | 0.662[acd] | 0.411[acd] | 0.601[acd] | 0.390[acd] | 0.666[acd] | 0.450[acd] | 0.607[acd] | 0.400[acd] | 0.467 | 0.520[cd] |
| mMSE | CE | 0.676[abd] | 0.422[abd] | 0.724[abd] | 0.484[abd] | 0.668[abd] | 0.448[abd] | 0.737[abd] | 0.511[abd] | 0.473 | 0.523[bd] |
| mMSE | Ensemble | 0.683[abc] | 0.414[abc] | 0.717[abc] | 0.482[abc] | 0.661[abc] | 0.458[abc] | 0.736[abc] | 0.516[abc] | 0.492 | 0.523[bc] |
| KL | Random | 0.571 | 0.361 | 0.661[bcd] | 0.428[bcd] | 0.529 | 0.356 | 0.625 | 0.416 | 0.447 | 0.511[c] |
| KL | BM25 | 0.655[cd] | 0.401[cd] | 0.698[acd] | 0.471[acd] | 0.637[cd] | 0.430[cd] | 0.726[cd] | 0.508[cd] | 0.466[c] | 0.504[d] |
| KL | CE | 0.660[bd] | 0.401[bd] | 0.712[abd] | 0.477[abd] | 0.633[bd] | 0.434[bd] | 0.728[bd] | 0.509[bd] | 0.467[b] | 0.513[a] |
| KL | Ensemble | 0.660[bc] | 0.402[bc] | 0.727[abc] | 0.494[abc] | 0.670[bc] | 0.455[bc] | 0.733[bc] | 0.519[bc] | 0.485 | 0.463[b] |

Table 2: Empirical Values of teacher entropy, the relative density ratio of each sampling domain and empirical measures of the query diameter. Each is taken at the $95^{\text{th}}$ percentile (either max or mean) for robust estimates. Entropy is measured in Nats via Shannon entropy over each ranking. We note that these are approximations of the theoretical aspects discussed in this work.

| Source | $\widehat{H}_\nu(g)$ | $\widehat{\kappa}_Q$ | $\widehat{\Delta}_Q$ |
|---|---|---|---|
| Random | 6.62±0.127 | 14.202±556.251 | 10.448±0.044 |
| BM25 | 4.973±1.978 | 12.747±461.588 | 9.862±0.205 |
| Cross-Encoder | 4.068±0.930 | 11.116±353.018 | 9.593±0.251 |
| Ensemble | 3.973±0.838 | 8.276±165.234 | 9.546±0.233 |

within the sampled teacher entropy distribution by quartile. In Table 3, we see that once locality is established, one can further improve effectiveness in-domain by selecting examples conditioned on ranking entropy. Even under a constrastive setting we find that sub-selection by some teacher (significantly less expensive then ensembling before similarly filtering) can further improve performance reducing the gap observed in our main results under a contrastive objective. We see that choosing the central mass of the entropy distribution (inner quartiles) is most effective and can be contrasted with selection in the outlier quartiles in which effectiveness degrades. Generally we observe that the upper quartile will yield greater effectiveness over the lower quartile, coupled with the reduced effectiveness of selection via outlier quartiles we consider that a balance must be struck between capturing high entropy examples for the purposes of generalisation. We note that all cases degrade out-of-domain potentially suggesting that choosing examples by these criteria induce overfitting to the particular cases within the training domain. We observe that increased mean entropy leads to reduced effectiveness, considering Table 2, we can infer that in cases where entropy is high, the representation space is insufficiently tight to compensate for this entropy. Out-of-domain correlation is minimal as the density ratio $\kappa_Q$ will inflate the VC term, bounding generalisation under this setting, thus any attempt to improve locality will fail to improve effectiveness.

**Differences in Model Behaviour** Even under densely annotated ($>$ 6 relevant texts per query (Craswell et al., 2020b;a)) test collections, variations in in-domain effectiveness between operational settings are minimal; this is expected via our bound. However, when observing intrinsic model behaviour, we observe that the chosen domain can greatly affect score distributions under otherwise identical optimization settings. In Figure 2, observe how different settings align with a power law; this can be considered alignment with a Zipfian distribution. See how optimization criteria lead to a vastly different score distribution from the teacher in Figure 2a, the original teacher has high

Table 3: IR effectiveness across domain subsets by quartiles (Q) of the teacher entropy distribution over training examples. Effectiveness is measured and evaluated as noted in Table 1. Shannon Entropy is denoted $\widehat{H}_\nu(g)$.

| Loss | Transform | $\widehat{H}_\nu(g)$ | TREC DL'19 | | TREC DL'20 | | BEIR |
|------|-----------|------|------|------|------|------|------|
| | | | nDCG | MAP | nDCG | MAP | nDCG |
| LCE | Original | 4.973±1.978 | 0.681 | 0.459 | 0.686 | 0.488 | 0.472 |
| LCE | Lower Q | 5.814±1.483 | 0.690 | 0.469 | 0.720 | 0.506 | 0.454 |
| LCE | Inner Qs | 5.344±1.527 | 0.723 | 0.491 | 0.742 | 0.520 | 0.465 |
| LCE | Upper Q | 5.106±1.027 | 0.716 | 0.482 | 0.739 | 0.518 | 0.460 |
| LCE | Outlier Qs | 6.807±1.278 | 0.638 | 0.412 | 0.636 | 0.425 | 0.357 |
| mMSE | Original | 4.973±1.978 | 0.601 | 0.390 | 0.607 | 0.400 | 0.520 |
| mMSE | Lower Q | 5.814±1.483 | 0.720 | 0.485 | 0.729 | 0.511 | 0.484 |
| mMSE | Inner Qs | 5.344±1.527 | 0.727 | 0.492 | 0.729 | 0.505 | 0.490 |
| mMSE | Upper Q | 5.106±1.027 | 0.724 | 0.491 | 0.737 | 0.504 | 0.491 |
| mMSE | Outlier Qs | 6.807±1.278 | 0.712 | 0.473 | 0.732 | 0.501 | 0.469 |

confidence within the top-10 ranks, and scores reach an elbow point at rank 218. However, depending on the sampling domain, we observe collapse when applying a random distribution in Figure 2b as both entropy and the relative density ratio, as outlined in Table 2, are insufficiently tight, leading to collapse. Conversely, we observe power law behaviour when applying an ensemble with an elbow at rank 12. Given the weak performance of ensemble approaches out-of-domain when this behaviour is present, we posit that this highly confident behaviour may be overfitting to an in-domain setting and may not be desirable. Though we leave any causal analysis to future work, we observe this behavior in several settings as shown in Appendix G.

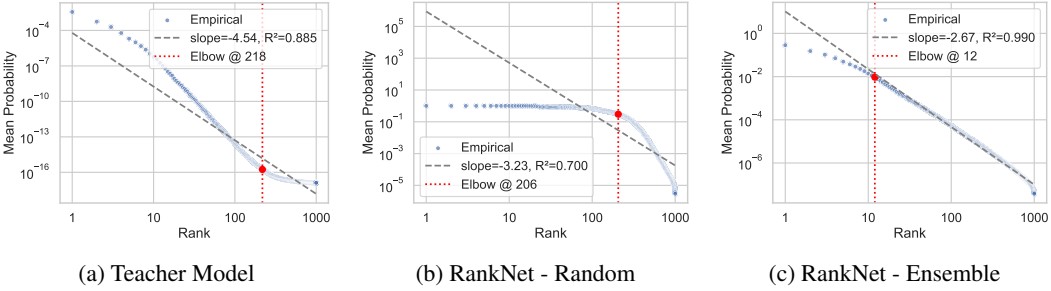

(a) Teacher Model  (b) RankNet - Random  (c) RankNet - Ensemble

Figure 2: Score versus rank ratio comparing the teacher (LCE) and two RankNet-trained students with different sampling distributions evaluated on TREC DL'19. Note the log-log scale and alignment with power laws plotted in grey.

## 4 CONCLUSION

In this work, we have provided a systematic analysis of two core components in modern neural ranking pipelines—example locality (negative sampling) and target entropy (distillation)—both theoretically and empirically. Our analysis establishes a novel generalisation bound for ranking distillation in terms of locality and entropy, accounting for biased sampling strategies. This bound shows that overly "hard" or overly uniform teacher distributions can both degrade student performance, and that geometry-driven sampling impacts only the bias term, not the entropy term. Empirically, across both in-domain (TREC Deep Learning 2019/2020) and out-of-domain (BEIR) benchmarks, we demonstrate that complex, multi-stage hard-negative pipelines yield minimal gains over simpler sampling strategies under distillation and theoretically justify cases where it is valuable to expend such computation. Furthermore, by stratifying examples according to teacher ranking entropy, we observe consistent in-domain improvements at intermediate entropy levels, while high-entropy "outlier" subsets degrade performance, confirming the bound's prediction. Moving forward, our findings encourage a shift away from computationally intensive ensemble and cascade architectures. By focusing on the two orthogonal dimensions we have identified, tangible improvements can be made in ranking effectiveness without excessive computational expense.

## 5 REPRODUCIBILITY STATEMENT

We have made several efforts to facilitate the reproducibility of our work, including the use of open models, training processes from the literature, and commonly available benchmarks. Parameter and training decisions were primarily motivated by well-known prior art as noted in Section 3.1. All data is publicly available with further details and licenses stated in Appendix E.1. Assumptions and full formal definitions and proofs for all theoretical discussions are described in Appendix A, B and C. Finally, our source code is provided in supplementary materials attached to this submission.

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

Table 4: Unified notation (evaluation under $\mu_Q$; training under $\pi_Q$).

| Symbol | Type | Domain | Meaning |
|---|---|---|---|
| $\mu_Q$ | measure | $\mathcal{X}$ | Reference/evaluation query measure. |
| $\pi_Q$ | measure | $\mathcal{X}$ | Training policy (unbiased if $\pi_Q = \mu_Q$). |
| $w_Q$ | weight | $(0, \infty)$ | Importance ratio $d\mu_Q/d\pi_Q$. |
| $w_Q^{(2)}$ | weight | $(0, \infty)$ | Pairwise importance ratio $w_Q(x)w_Q(x') = d\mu_Q^{\otimes 2}/d\pi_Q^{\otimes 2}$. |
| $\kappa_Q^{(2)}$ | scalar | $[1, \infty)$ | $\operatorname{ess\,sup} w_Q^{(2)} = \kappa_Q^2$. |
| $\kappa_Q$ | scalar | $[1, \infty)$ | $\operatorname{ess\,sup} w_Q$. |
| $d(\cdot, \cdot)$ | metric | $\mathcal{X}^2 \to \mathbb{R}_{\geq 0}$ | Ground metric. |
| $\Delta_Q$ | scalar | $[0, \infty)$ | Essential diameter under $\mu_Q^{\otimes 2}$. |
| $f, g, f^\star, \widehat{f}$ | scorers | $\mathcal{X} \to \mathbb{R}$ | Student, teacher, target, ERM. |
| $\delta_{x,x'}(h)$ | diff. | $\mathbb{R}$ | $h(x) - h(x')$. |
| $D_\phi$ | divergence | $\mathbb{R}_{\geq 0}$ | Bregman divergence from potential $\phi$. |
| $Z_h$ | surrogate | $\mathbb{R}_{\geq 0}$ | $D_\phi(\delta(h)\|\delta(g))$. |
| $H(g)$ | entropy | $[0, \log 2]$ | Pairwise entropy under $\mu_Q^{\otimes 2}$. |
| $\eta$ | function | $[0, \log 2] \to [0, \frac{1}{2}]$ | $\eta(h) = \frac{1}{2} - \sqrt{(\log 2 - h)/2}$. |
| $d$ | capacity | $\mathbb{N}$ | VC dim. of $\{\mathbf{1}[Z_h > \tau]\}$. |
| $n, \delta$ | scalars | – | Sample size; confidence level. |
| $L$ | scalar | $(0, \infty)$ | Lipschitz constant of $f^\star$. |
| $\zeta$ | scalar | $(0, \infty)$ | Calibration constant (e.g. $\sqrt{2}$ for RankNet). |

## A  PRELIMINARIES AND DEFINITIONS

For completeness, we collect all loss functions used in the main text and show that each can be expressed as a *Bregman divergence* (Bregman, 1967). We recall the definition first.

**Definition A.1** (Bregman divergence). *Let $\phi : \mathbb{R} \to \mathbb{R}$ be strictly convex and $C^1$. The Bregman divergence between $a, b \in \mathbb{R}$ is*

$$D_\phi(a\|b) = \phi(a) - \phi(b) - \phi'(b)(a - b).$$

### A.1  RANKNET AS A BREGMAN DIVERGENCE

Let $s_{ij} = f(\mathcal{X}_i) - f(\mathcal{X}_j)$ and $p_{ij}(f) = \sigma(s_{ij})$ with $\sigma(z) = 1/(1 + e^{-z})$. For pair labels $y_{ij} \in \{0, 1\}$ the RankNet loss is

$$\mathcal{L}_{\text{RankNet}}(f) = -\big[y_{ij} \log p_{ij}(f) + (1 - y_{ij}) \log(1 - p_{ij}(f))\big].$$

Let $\phi(u) = u \log u + (1 - u) \log(1 - u)$ on $u \in (0, 1)$ (negative binary entropy). See also Banerjee et al. (2005) for the connection between Bregman divergences and exponential families. Its Bregman divergence equals the Bernoulli KL:

$$D_\phi(a\|b) = a \log \frac{a}{b} + (1 - a) \log \frac{1 - a}{1 - b}.$$

For $y_{ij} \in \{0, 1\}$ this reduces to the negative log-likelihood above. Hence

$$\mathcal{L}_{\text{RankNet}}(f) = D_\phi\big(y_{ij} \,\|\, p_{ij}(f)\big).$$

### A.2  MARGINMSE AS A BREGMAN DIVERGENCE

Let the student/teacher margins be $m_{ij}(f) = f(\mathcal{X}_i) - f(\mathcal{X}_j)$ and $m_{ij}(g) = g(\mathcal{X}_i) - g(\mathcal{X}_j)$. For $\phi(u) = u^2$, the Bregman divergence is $D_\phi(a\|b) = (a - b)^2$, so

$$\mathcal{L}_{\text{MarginMSE}}(f) = D_\phi\big(m_{ij}(f) \,\|\, m_{ij}(g)\big).$$

## A.3 DEFINITIONS

**Definition A.2** (Lipschitz continuity (Weaver, 1999)). *A function $h : (\mathcal{X}, d) \to \mathbb{R}$ is $L$-Lipschitz if there exists $L > 0$ such that*

$$|h(x) - h(x')| \le L \cdot d(x, x') \quad \text{for all } x, x' \in \mathcal{X}.$$

**Definition A.3** (Pair-wise risk under a reference measure). *For a scorer $f$ and reference query measure $\mu_Q$,*

$$\mathcal{R}_{\mu_Q}(f) := \Pr_{(x,x') \sim \mu_Q^{\otimes 2}} \big[ f(x) < f(x') \big] = \mathbb{E}_{\mu_Q^{\otimes 2}} \big[ \mathbf{1}\{f(x) < f(x')\} \big].$$

**Definition A.4** (Sampling policy, importance weights, and condition numbers). *Training pairs are drawn from a (possibly biased) policy $\pi_Q$ as $(x, x') \sim \pi_Q^{\otimes 2}$; risk is always evaluated under $\mu_Q$. Define the single-point Radon–Nikodym ratio and its essential supremum*

$$w_Q(x) = \frac{d\mu_Q}{d\pi_Q}(x), \qquad \kappa_Q = \operatorname*{ess\,sup}_{x \in \operatorname{supp} \pi_Q} w_Q(x) \in [1, \infty),$$

*and the pairwise ratio*

$$w_Q^{(2)}(x, x') = \frac{d\mu_Q^{\otimes 2}}{d\pi_Q^{\otimes 2}}(x, x') = w_Q(x)\, w_Q(x'), \qquad \kappa_Q^{(2)} = \operatorname*{ess\,sup}_{(x,x') \in \operatorname{supp} \pi_Q^{\otimes 2}} w_Q^{(2)}(x, x') = \kappa_Q^2.$$

*When $\pi_Q = \mu_Q$ (unbiased sampling), $w_Q \equiv 1$ so $\kappa_Q = 1$ and $\kappa_Q^{(2)} = 1$.*

**Definition A.5** (VC dimension for function classes). *For a hypothesis class $\mathcal{H} \subseteq \mathbb{R}^{\mathcal{X}}$, Consider the induced thresholded classes $\{x \mapsto \mathbf{1}[h(x) > t] : h \in \mathcal{H}, t \in \mathbb{R}\}$. The VC dimension of $\mathcal{H}$ is the VC dimension of this induced class of indicator functions.*

**Definition A.6** (Essential diameter). *For a probability space $(\mathcal{X}, \mu_Q)$ with metric $d$, the essential diameter is*

$$\Delta_Q = \inf\{M \ge 0 : \mu_Q^{\otimes 2}(\{(x, x') : d(x, x') > M\}) = 0\}.$$

**Definition A.7** (Pair-wise entropy). *For a scorer $g : \mathcal{X} \to \mathbb{R}$ and measure $\mu_Q$, let $p_{x,x'} := \Pr[g(x) > g(x')]$. The pair-wise entropy is*

$$H(g) = -\mathbb{E}_{(x,x') \sim \mu_Q^{\otimes 2}} \big[ p_{x,x'} \log p_{x,x'} + (1 - p_{x,x'}) \log(1 - p_{x,x'}) \big].$$

# B BOUNDING MISORDERING UNDER TEACHER TARGETS

**Lemma B.1** (Pinsker bound from pairwise entropy to misordering). *Fix a pair $(x, x')$ and let $Z = \mathbf{1}\{g(x) > g(x')\}$, $p = \Pr[Z = 1]$, $P = (p, 1 - p)$, and $U = (\frac{1}{2}, \frac{1}{2})$. Then*

$$\epsilon(p) := \Pr[g(x) < g(x')] = \min(p, 1 - p) \ge \eta(H(p)) := \frac{1}{2} - \sqrt{\frac{\log 2 - H(p)}{2}},$$

*where $H(p) = -p \log p - (1 - p) \log(1 - p)$ (nats). Averaging over $(x, x') \sim \mu_Q^{\otimes 2}$ gives*

$$\mathbb{E}_{(x,x')} \Pr\{g \text{ misorders } (x, x')\} \ge \eta(H(g)), \quad H(g) := \mathbb{E}_{(x,x')} H(p_{x,x'}).$$

*Proof.* By Pinsker following Painsky & Wornell (2020), $\|P - U\|_{\mathrm{TV}} \le \sqrt{\frac{1}{2} D_{\mathrm{KL}}(P \| U)}$. Explicitly,

$$D_{\mathrm{KL}}(P \| U) = p \log \frac{p}{1/2} + (1 - p) \log \frac{1 - p}{1/2} = -H(p) + \log 2.$$

Moreover, $\|P - U\|_{\mathrm{TV}} = |p - \frac{1}{2}|$. Hence $|p - \frac{1}{2}| \le \sqrt{(\log 2 - H(p))/2}$. Since $\epsilon(p) = \min(p, 1 - p) = \frac{1}{2} - |p - \frac{1}{2}|$, we have

$$\epsilon(p) \ge \frac{1}{2} - \sqrt{\frac{\log 2 - H(p)}{2}} = \eta(H(p)).$$

For the population statement, write $p_{x,x'} = \Pr[Z = 1 \mid x, x']$ and use the concavity of $H$ and convexity of $\eta \circ H$:

$$\mathbb{E}\,\epsilon(p_{x,x'}) \ge \mathbb{E}\,\eta(H(p_{x,x'})) \ge \eta(\mathbb{E}\,H(p_{x,x'})) = \eta(H(g)).$$

$\square$

**Lemma B.2** (Uniform deviation for bounded surrogate class). *Let $\mathcal{H} \subseteq \mathbb{R}^{\mathcal{X}}$ and define $Z_h(x, x') := D_\phi(\delta_{x,x'}(h)\|\delta_{x,x'}(g)) \in [0, B]$ for all $(x, x')$ under $\mu_Q^{\otimes 2}$, with $B = L\Delta_Q$. Assume the thresholded class $\{(x, x', \tau) \mapsto \mathbf{1}[Z_h(x, x') > \tau] : h \in \mathcal{H}\}$ has VC dimension $d < \infty$ in $(x, x')$ uniformly over $\tau$. For i.i.d. $(x_s, x'_s) \sim \mu_Q^{\otimes 2}$, any $\delta \in (0, 1)$, with probability $\geq 1 - \delta$,*

$$\sup_{h \in \mathcal{H}} \left| \tfrac{1}{n} \sum_{s=1}^{n} Z_h(x_s, x'_s) - \mathbb{E}Z_h \right| \leq B \sqrt{\tfrac{2}{n} \left( d \log \tfrac{2en}{d} + \log \tfrac{4}{\delta} \right)}.$$

*Proof.* By symmetrisation for bounded real-valued classes, for i.i.d. samples and i.i.d. Rademacher signs $\epsilon_s$,

$$\mathbb{E} \sup_h \left| \tfrac{1}{n} \sum_s (Z_h - \mathbb{E}Z_h) \right| \leq 2 \mathbb{E} \sup_h \left| \tfrac{1}{n} \sum_s \epsilon_s Z_h(x_s, x'_s) \right|.$$

Since $Z_h \in [0, B]$, apply the standard contraction via integration over thresholds:

$$Z_h(x, x') = \int_0^B \mathbf{1}\{Z_h(x, x') > \tau\} \, d\tau,$$

so

$$\sup_h \left| \tfrac{1}{n} \sum_s \epsilon_s Z_h(x_s, x'_s) \right| \leq \int_0^B \sup_h \left| \tfrac{1}{n} \sum_s \epsilon_s \mathbf{1}\{Z_h(x_s, x'_s) > \tau\} \right| d\tau.$$

For each fixed $\tau$, $\mathcal{C}_\tau := \{\mathbf{1}[Z_h(\cdot, \cdot) > \tau] : h \in \mathcal{H}\}$ is a class of $\{0, 1\}$-valued functions with VC dimension $\leq d$ by assumption; hence its empirical Rademacher average is bounded by $\sqrt{\tfrac{2}{n} \log N(2n, d)} \leq \sqrt{\tfrac{2}{n} d \log(\tfrac{2en}{d})}$ using the growth function bound $\sum_{k=0}^{d} \binom{n}{k} \leq (en/d)^d$ via Sauer's lemma (Sauer, 1972)). Therefore,

$$\mathbb{E} \sup_h \left| \tfrac{1}{n} \sum_s \epsilon_s Z_h(x_s, x'_s) \right| \leq \int_0^B \sqrt{\tfrac{2}{n} d \log(\tfrac{2en}{d})} \, d\tau = B \sqrt{\tfrac{2}{n} d \log(\tfrac{2en}{d})}.$$

Combining with symmetrisation yields

$$\mathbb{E} \sup_h \left| \tfrac{1}{n} \sum_s (Z_h - \mathbb{E}Z_h) \right| \leq 2B \sqrt{\tfrac{2}{n} d \log(\tfrac{2en}{d})}.$$

A bounded-difference concentration (McDiarmid) or Bernstein then upgrades expectation to high probability: for any $\delta \in (0, 1)$, with probability $\geq 1 - \delta$,

$$\sup_h \left| \tfrac{1}{n} \sum_s (Z_h - \mathbb{E}Z_h) \right| \leq B \sqrt{\tfrac{2}{n} \left( d \log \tfrac{2en}{d} + \log \tfrac{4}{\delta} \right)}.$$

$\square$

**Lemma B.3** (Calibration of RankNet to pairwise 0–1 risk). *Let $p^\star(x, x') := \Pr\{Y = 1 \mid x, x'\}$ with $Y = \mathbf{1}\{f^\star(x) < f^\star(x')\}$, and let $p_f(x, x') := \sigma(f(x) - f(x'))$. Define the (conditional) logistic excess $\Delta_{\log}(x, x') := \mathrm{KL}(\mathrm{Bern}(p^\star)\|\mathrm{Bern}(p_f))$. Then the pairwise 0–1 excess risk satisfies*

$$\Pr\{f(x) < f(x')\} - \Pr\{f^\star(x) < f^\star(x')\} \leq \sqrt{2 \, \Delta_{\log}(x, x')}.$$

*Consequently,*

$$\mathcal{R}_{\mu_Q}(f) - \mathcal{R}_{\mu_Q}(f^\star) \leq \sqrt{2 \, \mathbb{E}_{\mu_Q^{\otimes 2}} \Delta_{\log}(x, x')} \leq \sqrt{2 \, \mathbb{E}_{\mu_Q^{\otimes 2}} D_\phi(Y \| p_f)},$$

*so for RankNet we can take $\zeta = \sqrt{2}$ in the main theorems.*

*Proof.* Fix $(x, x')$ and abbreviate $p^\star = p^\star(x, x')$, $q = p_f(x, x')$, $U = (\tfrac{1}{2}, \tfrac{1}{2})$. The Bayes optimal decision minimises 0–1 error by predicting $\mathbf{1}[p^\star \geq \tfrac{1}{2}]$; the 0–1 excess at $(x, x')$ equals $|p^\star - \tfrac{1}{2}| - \mathbf{1}[p^\star \geq \tfrac{1}{2}](p^\star - \tfrac{1}{2}) = |p^\star - q^\star|$ evaluated at the sign boundary and is upper bounded by the total variation $\|\mathrm{Bern}(p^\star) - \mathrm{Bern}(q)\|_{\mathrm{TV}} = |p^\star - q|$. Pinsker gives $|p^\star - q| \leq \sqrt{\tfrac{1}{2} \mathrm{KL}(\mathrm{Bern}(p^\star)\|\mathrm{Bern}(q))}$. Thus the conditional 0–1 excess is $\leq \sqrt{2 \, \Delta_{\log}(x, x')}$. Apply Jensen to move the square root outside the expectation:

$$\mathbb{E}|p^\star - q| \leq \mathbb{E}\sqrt{2 \, \Delta_{\log}} \leq \sqrt{2 \, \mathbb{E}\Delta_{\log}}.$$

Finally, note $\Delta_{\log} = D_\phi(Y \| q)$ when $Y \in \{0, 1\}$ and $\phi(u) = u \log u + (1 - u) \log(1 - u)$ as in RankNet. $\square$

## C   GENERALISATION UNDER RISK FROM TEACHER ENTROPY AND LOCALITY

**Theorem C.1** (Locality–Entropy excess risk under a unified sampling policy). *Assume: (i) $f^\star$ is L-Lipschitz on $(\mathcal{X}, d)$ and $\Delta_Q := \operatorname{ess\,sup}_{(x,x')\sim\mu_Q^{\otimes 2}} d(x,x') < \infty$; (ii) teacher $g$ induces pairwise entropy $H(g)$ as in Lemma B.1; (iii) the surrogate $Z_h(x,x') := D_\phi(\delta_{x,x'}(h)\|\delta_{x,x'}(g))$ is bounded by $L\Delta_Q$ and $h \mapsto Z_h$ is 1–Lipschitz in the score difference; (iv) $\widehat{f}$ minimises the empirical (possibly importance-weighted) surrogate over $n$ pairs drawn i.i.d. from $\pi_Q^{\otimes 2}$ and we denote $w_Q = d\mu_Q/d\pi_Q$, $\kappa_Q = \operatorname{ess\,sup} w_Q$. Then, for any $\delta \in (0,1)$, with probability at least $1 - \delta$,*

$$\mathcal{R}_{\mu_Q}(\widehat{f}) - \mathcal{R}_{\mu_Q}(f^\star) \;\leq\; \zeta\, L\Delta_Q\, \eta\big(H(g)\big) \;+\; C\,\kappa_Q\,\sqrt{\frac{d\log(1/\delta)}{n}}.$$

*For RankNet (logistic surrogate), one may set $\zeta = \sqrt{2}$ by Lemma B.3. In the unbiased case ($\pi_Q = \mu_Q$) we have $\kappa_Q = 1$.*

*Proof.* **Step 1 (Teacher–Bayes gap via entropy and locality).** By Lipschitzness of $f^\star$, $|f^\star(x) - f^\star(x')| \leq Ld(x,x') \leq L\Delta_Q$. Lemma B.1 yields a lower bound on the teacher's misordering rate in terms of $H(g)$; thus deviations of $g$ from the Bayes order contribute at most $L\Delta_Q\, \eta(H(g))$ to the risk gap (the geometry scales score separations, entropy controls sign errors).

**Step 2 (Calibration of surrogate to 0–1 pairwise risk).** By classification calibration, there exists $\zeta > 0$ such that

$$\mathcal{R}_{\mu_Q}(h) - \mathcal{R}_{\mu_Q}(g) \;\leq\; \zeta\, \mathbb{E}_{\mu_Q^{\otimes 2}} Z_h \quad \text{for all } h \in \mathcal{H}.$$

For RankNet this holds with $\zeta = \sqrt{2}$ by Lemma B.3; for other $D_\phi$ one may use standard composite/proper loss calibration (constant absorbed into $\zeta$).

**Step 3 (Population envelope and Lipschitz control).** By assumption (iii), $0 \leq Z_h \leq L\Delta_Q$ for all $(x,x')$ and $|Z_h - Z_{f^\star}| \leq |\delta(h) - \delta(f^\star)| \leq 2Ld(x,x') \leq 2L\Delta_Q$, so $\mathbb{E} Z_{f^\star} \leq L\Delta_Q$.

**Step 4 (Weighted uniform deviation under $\pi_Q$).** Consider the weighted empirical process $\frac{1}{n}\sum_{s=1}^n w_Q^{(2)}(x_s, x_s')\, Z_h(x_s, x_s')$ with $(x_s, x_s') \sim \pi_Q^{\otimes 2}$. Because $0 < w_Q^{(2)} \leq \kappa_Q^{(2)} = \kappa_Q^2$ a.s. and $Z_h \in [0, L\Delta_Q]$, the envelope is bounded by $\kappa_Q^2 L\Delta_Q$. Applying Lemma B.2 with $B = \kappa_Q^2 L\Delta_Q$ and using $\sqrt{\kappa_Q^{(2)}} = \kappa_Q$ in the resulting rate gives, with probability $\geq 1 - \delta$,

$$\sup_h \Big| \mathbb{E}_{\mu_Q^{\otimes 2}} Z_h - \frac{1}{n}\sum_s w_Q^{(2)}(x_s, x_s')\, Z_h(x_s, x_s') \Big| \;\leq\; C_1\, L\Delta_Q\, \kappa_Q\, \sqrt{\frac{d\log(1/\delta)}{n}}.$$

**Step 5 (ERM inequality and cancellation).** By empirical optimality of $\widehat{f}$, $\frac{1}{n}\sum_s w_Q^{(2)} Z_{\widehat{f}} \leq \frac{1}{n}\sum_s w_Q^{(2)} Z_{f^\star}\dots$ hence

$$\mathbb{E}_{\mu_Q^{\otimes 2}} Z_{\widehat{f}} - \mathbb{E}_{\mu_Q^{\otimes 2}} Z_{f^\star} \;\leq\; 2C_1\, L\Delta_Q\, \kappa_Q\, \sqrt{\frac{d\log(1/\delta)}{n}}.$$

Subtract the two deviations from Step 4 (once with $h = \widehat{f}$, once with $h = f^\star$) and use the ERM inequality to get

$$\mathbb{E}_{\mu_Q^{\otimes 2}} Z_{\widehat{f}} - \mathbb{E}_{\mu_Q^{\otimes 2}} Z_{f^\star} \;\leq\; 2C_1\, L\Delta_Q \sqrt{\frac{\kappa_Q\, d\log(1/\delta)}{n}}.$$

Absorb the factor 2 into $C := 2C_1$.

**Step 6 (Assemble).** Decompose

$$\mathcal{R}_{\mu_Q}(\widehat{f}) - \mathcal{R}_{\mu_Q}(f^\star) = \big[\mathcal{R}_{\mu_Q}(\widehat{f}) - \mathcal{R}_{\mu_Q}(g)\big] + \big[\mathcal{R}_{\mu_Q}(g) - \mathcal{R}_{\mu_Q}(f^\star)\big].$$

Bound the first bracket by calibration (Step 2) and the second by locality–entropy (Step 1); replace $\mathbb{E} Z_{\widehat{f}}$ by $\mathbb{E} Z_{f^\star}$ plus the deviation from Step 5 and use $\mathbb{E} Z_{f^\star} \leq L\Delta_Q$ (Step 3). This yields

$$\mathcal{R}_{\mu_Q}(\widehat{f}) - \mathcal{R}_{\mu_Q}(f^\star) \;\leq\; \zeta\, L\Delta_Q\, \eta(H(g)) \;+\; C\, L\Delta_Q \sqrt{\frac{\kappa_Q\, d\log(1/\delta)}{n}},$$

and we absorb $L\Delta_Q$ into $C$ in the statement if desired. $\qquad\qquad\square$

If the teacher is not degenerate, i.e. $\eta(H(g)) \geq \varepsilon > 0$, then $\zeta L\Delta_Q \leq \zeta L\Delta_Q \varepsilon^{-1}\eta(H(g))$ and the first two bias terms merge, absorbing all numerical constants into a single $C$ gives the form we provide in the main body of this work, this reduced form is realistic given the nature of ranking model teachers in which degenerate cases are rare due to the smooth estimations provided by neural models through activation functions such as softmax.

## D  BIASED-SAMPLING UNDER NEGATIVE MINERS

**Corollary D.1** (Biased sampling via importance weights). *Let $\pi_Q$ be any retrieval/miner policy with $\operatorname{supp} \pi_Q \subseteq \operatorname{supp} \mu_Q$ and $\kappa_Q = \operatorname{ess\,sup}(d\mu_Q/d\pi_Q) < \infty$. If $\widehat{f}$ minimises the importance-weighted empirical surrogate built from $n$ pairs drawn i.i.d. from $\pi_Q^{\otimes 2}$, then the bound of Theorem C.1 holds with this $\kappa_Q$ following importance-weighted risk under covariate shift (Hsu et al., 2021)". In particular,*

$$\mathcal{R}_{\mu_Q}(\widehat{f}) - \mathcal{R}_{\mu_Q}(f^\star) \leq \zeta L\Delta_Q \eta(H(g)) + C \kappa_Q \sqrt{\frac{d\log(1/\delta)}{n}}.$$

*Proof.* Repeat Steps 4–6 of Theorem C.1 with the weighted process $w_Q^{(2)} Z_h$; the only change is the envelope $B = \kappa_Q L\Delta_Q$ in Lemma B.2, which introduces the factor $\kappa_Q$ in the rate. All other steps are unchanged. □

## E  ADDITIONAL EXPERIMENTAL DETAILS

### E.1  DATASET DESCRIPTIONS

Table 5 describes all test collections in terms of their domain, queries and corpus size. In all cases we rerank BM25 ($k_1 = 1.2$, $b = 0.75$). We found in further experimentation that due to the point-wise nature of models trained in this investigation, biases remained consistent under different re-rankers thus for conciseness we solely present BM25.

Table 5: Descriptive statistics for all test collections, $|\mathcal{Q}|$ indicates the number of test queries, $|\mathcal{D}|$ indicates the corpus size used in retrieval and ranking. In all cases, we re-rank BM25.

| Dataset | Domain | $|\mathcal{Q}|$ | $|\mathcal{D}|$ |
|---|---|---|---|
| TREC Deep-Learning 2019 | Ad-Hoc Web Search | 43 | 8E6 |
| TREC Deep-Learning 2020 | Ad-Hoc Web Search | 53 | |
| ArguAna | Argument Retrieval | 1406 | 8.67E3 |
| Climate-Fever | Environmental | 1535 | 542E3 |
| CQADupStack | OpenQA | 13145 | 457E3 |
| DBPedia | OpenQA | 400 | 463E3 |
| FiQA | OpenQA | 648 | 57E3 |
| HotpotQA | OpenQA | 7405 | 523E3 |
| NFCorpus | Medical | 323 | 36E2 |
| NQ | OpenQA | 3452 | 268E3 |
| Quora | OpenQA | 10000 | 523E3 |
| SCIDOCS | Academic | 1000 | 25E3 |
| SciFact | Academic | 300 | 5E3 |
| TREC Covid | Medical | 50 | 171E3 |
| Touche 2020 | Argument Retrieval | 49 | 382E3 |

### E.2  EMPIRICAL APPROXIMATIONS

We apply Monte-Carlo sampling to provide empirical estimates of theoretical values outlined in Section 2, our sample size is equal to our training corpus (maximising possible samples under our

setting). We compute $H_\nu(g)$ over teacher scores of training data observed during the training of each model.

As our true measure $\mu_Q$ over $\mathcal{X}$ is latent (or infeasible to compute with standard benchmarks due to query mismatch), we provide an approximation of $\kappa_Q$, the density ratio bridging our biased measure $\nu_Q$ to our unbiased risk. We do so by assuming a uniform chance of sampling all documents as opposed to a rank-biased sample taking $1/g(Q, D), \forall D \in \mathcal{X}_Q$, $\kappa_Q$ is then taken as the supremum of these values.

To compute an empirical diameter $\widehat{\Delta}_Q$, we apply cosine distance over representation from an existing embedding model (we use RetroMAE (Xiao et al., 2022), a strong embedding model based on MAE pre-training) as our measure $d$ and compute the supremum over Monte-Carlo samples from our training data over each query.

### E.3 TOST test

A two one-sided t-test (TOST) determines if the means of two populations are equivalent based on independent samples from each population, in our case, the query-level effectiveness of two models. For means $\mu_1, \mu_2$ and confidence bound $\theta = |\mu_2 - \mu_1| \cdot \epsilon$ ($\epsilon$ is a percentage bound parameter), we assess two hypotheses, that $\mu_2 - \mu_1$ lies above $\theta$ and below $\theta$ using one-sided t-tests with confidence $1 - 2\alpha$ compensating for multiple hypothesis testing. Thus, within an $\epsilon$ bound with confidence $1 - \alpha$, we can say that $\mu_1, \mu_2$ are statistically equivalent.

## F OUT-OF-DOMAIN EFFECTIVENESS

Table 6 shows all BEIR splits for bi-encoder models. Table 7 shows all splits for cross-encoder models.

Table 6: Mean nDCG@10 for architecture BE across BEIR datasets

| Loss | Domain | arguana | climate-fever | cqa-android | cqa-english | cqa-gaming | cqa-gis | cqa-mathematica | cqa-physics | cqa-programmers | cqa-stats | cqa-tex | cqa-unix | cqa-webmasters | cqa-wordpress | dbpedia-entity | fiqa | hotpotqa | nfcorpus | nq | quora | scidocs | scifact | trec-covid | webis-touche2020 |
|---|---|---|---|---|---|---|---|---|---|---|---|---|---|---|---|---|---|---|---|---|---|---|---|---|---|
| LCE | Random | 0.414 | 0.234 | 0.319^BCD | 0.295^C | 0.393^BC | 0.220^BC | 0.169^BC | 0.308 | 0.254^BC | 0.194^BCD | 0.182 | 0.230^C | 0.244^BC | 0.195^BCD | 0.318^BC | 0.241 | 0.547 | 0.296^BCD | 0.322 | 0.810 | 0.130 | 0.534^BCD | 0.628^BCD | 0.291^BCD |
| LCE | BM25 | 0.303 | 0.193 | 0.333^ACD | 0.307 | 0.398^AC | 0.215^AC | 0.163^AC | 0.284^C | 0.267^AC | 0.201^ACD | 0.201 | 0.241 | 0.254^AC | 0.217^ACD | 0.321^AC | 0.259^CD | 0.559 | 0.291^ACD | 0.407^D | 0.798 | 0.110 | 0.468^ACD | 0.671^ACD | 0.296^ACD |
| LCE | CE | 0.281 | 0.191 | 0.317^ABD | 0.292^A | 0.392^AB | 0.204^AB | 0.154^AB | 0.282^B | 0.255^AB | 0.191^ABD | 0.193 | 0.231^A | 0.260^AB | 0.205^ABD | 0.324^AB | 0.261^BD | 0.555 | 0.283^ABD | 0.405 | 0.791 | 0.104 | 0.491^ABD | 0.673^ABD | 0.284^ABD |
| LCE | Ensemble | 0.352 | 0.199 | 0.340^ABC | 0.327 | 0.427 | 0.237 | 0.183 | 0.321 | 0.282 | 0.208^ABC | 0.214 | 0.255 | 0.284 | 0.227^ABC | 0.346 | 0.272^BC | – | 0.303^ABD | 0.407^B | 0.823 | 0.120 | 0.507^ABC | 0.685^ABC | 0.336^ABC |
| RankNet | Random | 0.181 | 0.086 | 0.144 | 0.145 | 0.125 | 0.104 | 0.100 | 0.151 | 0.145 | 0.105 | 0.099 | 0.119 | 0.120 | 0.100 | 0.247 | 0.142 | 0.335 | 0.287^BCD | 0.238 | 0.707 | 0.097 | 0.259 | 0.593^BCD | 0.237 |
| RankNet | BM25 | 0.308 | 0.229 | 0.313 | 0.299 | 0.384 | 0.205 | 0.196 | 0.278 | 0.295^C | 0.195^C | 0.179 | 0.232^C | 0.243^C | 0.194^C | 0.354^CD | 0.274^CD | 0.592 | 0.293^CD | 0.415^D | 0.802 | 0.115 | 0.509^C | 0.657^ACD | 0.306^CD |
| RankNet | CE | 0.294 | 0.215 | 0.333^D | 0.310 | 0.402 | 0.219 | 0.168 | 0.291 | 0.256^B | 0.201^B | 0.188 | 0.238^B | 0.249^B | 0.207^B | 0.351^BD | 0.277^BD | 0.604 | 0.296^ABD | 0.420 | 0.807 | 0.113 | 0.502^B | 0.646^ABD | 0.323^BD |
| RankNet | Ensemble | 0.344 | 0.256 | 0.348^C | 0.338 | 0.444 | 0.251 | 0.185 | 0.324 | 0.281 | 0.224 | 0.212 | 0.265 | 0.274 | 0.226 | 0.363^BC | 0.282^BC | 0.599 | 0.310^ABC | 0.416^B | 0.835 | 0.131 | 0.554 | 0.681^ABC | 0.357^BC |
| mMSE | Random | 0.382 | 0.218^D | 0.324^C | 0.309^C | 0.393^C | 0.199^BC | 0.154^BC | 0.304^D | 0.257^BC | 0.182 | 0.172 | 0.212 | 0.265 | 0.184 | 0.351^D | 0.241 | 0.525 | 0.299^BCD | 0.384 | 0.818 | 0.117 | 0.498^BCD | 0.679^BCD | 0.314^BC |
| mMSE | BM25 | 0.299 | 0.193 | 0.293 | 0.295 | 0.363 | 0.200^AC | 0.156 | 0.270 | 0.254^AC | 0.199^CD | 0.184 | 0.229^AC | 0.239^AC | 0.202^C | 0.335^C | 0.262^C | 0.587 | 0.288^ACD | 0.414^C | 0.816 | 0.111 | 0.496^ACD | 0.662^ACD | 0.298^BCD |
| mMSE | CE | 0.312 | 0.190 | 0.323^A | 0.306^A | 0.402^A | 0.210^AB | 0.161^AB | 0.386 | 0.261^AB | 0.207^BD | 0.194 | 0.236^AB | 0.252^AB | 0.209^B | 0.332^B | 0.262^B | 0.591 | 0.293^ABD | 0.416^B | 0.618 | 0.108 | 0.504^ABD | 0.653^ABD | 0.313^ABD |
| mMSE | Ensemble | 0.366 | 0.219^A | 0.343 | 0.328 | 0.429 | 0.245 | 0.188 | 0.314^A | 0.282 | 0.216^BC | 0.217 | 0.262 | 0.284 | 0.231 | 0.351^A | 0.276 | 0.599 | 0.304^ABC | 0.407 | 0.840 | 0.121 | 0.537^ABC | 0.679^ABC | 0.349^ABC |
| KL | Random | 0.346 | 0.232 | 0.312^BCD | 0.308^BC | 0.405^BC | 0.218^BC | 0.164^BC | 0.302^BC | 0.259^BC | 0.197^BCD | 0.179 | 0.234^BC | 0.246^BCD | 0.201^BCD | 0.334^BCD | 0.262^BCD | 0.559 | 0.303^BCD | 0.359 | 0.745 | 0.132 | 0.539^CD | 0.666^BCD | 0.298^BCD |
| KL | BM25 | 0.324^C | 0.196 | 0.334^ACD | 0.315^AC | 0.403^AC | 0.214^AC | 0.163^AC | 0.267^AC | 0.267^AC | 0.202^ACD | 0.179 | 0.241^AC | 0.253^ACD | 0.215^ACD | 0.339^ACD | 0.275^ACD | 0.559 | 0.286^ACD | 0.414^D | 0.806 | 0.109 | 0.466 | 0.687^ACD | 0.322^ACD |
| KL | CE | 0.321^B | 0.207^D | 0.324^ABD | 0.303^AB | 0.407^AB | 0.209^AB | 0.160^AB | 0.301^AB | 0.258^AB | 0.204^ABD | 0.193 | 0.237^AB | 0.265^ABD | 0.210^ABD | 0.340^ABD | 0.268^ABD | 0.565 | 0.289^ABD | 0.413^ABD | 0.805 | 0.111 | 0.501^AD | 0.670^ABD | 0.323^ABC |
| KL | Ensemble | 0.354 | 0.206^C | 0.339^ABC | 0.333 | 0.431 | 0.238 | 0.184 | 0.321 | 0.281 | 0.216^ABC | 0.213 | 0.258 | 0.283^ABC | 0.228^ABC | 0.350^ABC | 0.277^ABC | 0.585 | 0.306^ABC | 0.413^ABC | 0.827 | 0.120 | 0.519^AC | 0.669^ABC | 0.344^ABC |

Table 7: Mean nDCG@10 for architecture CE across BEIR datasets

| Loss | Domain | arguana | climate-fever | cqa-android | cqa-english | cqa-gaming | cqa-gis | cqa-mathematica | cqa-physics | cqa-programmers | cqa-stats | cqa-tex | cqa-unix | cqa-webmasters | cqa-wordpress | dbpedia-entity | fiqa | hotpotqa | nfcorpus | nq | quora | scidocs | scifact | trec-covid | webis-touche2020 |
|---|---|---|---|---|---|---|---|---|---|---|---|---|---|---|---|---|---|---|---|---|---|---|---|---|---|
| LCE | Random | 0.311 | 0.217 | 0.386^D | 0.373 | 0.475 | 0.292 | 0.335^D | 0.306^D | 0.245^C | 0.288^BCD | 0.329 | 0.276^BCD | 0.369 | 0.317^C | 0.680 | 0.340^BCD | 0.384 | 0.795 | 0.156 | 0.665^BD | 0.658^BCD | 0.340^BCD |
| LCE | BM25 | 0.238 | 0.205 | 0.356^C | 0.318 | 0.436 | 0.253^C | 0.208^CD | 0.305 | 0.266^C | 0.232^C | 0.226^C | 0.288^ACD | 0.284^CD | 0.249^BCD | 0.404^CD | 0.361^C | 0.686 | 0.328^ACD | 0.474^CD | 0.670 | 0.147 | 0.643^AD | 0.728^ACD | 0.339^ACD |
| LCE | CE | 0.250 | 0.190 | 0.363^B | 0.298 | 0.447^D | 0.261^B | 0.198^ABD | 0.287 | 0.267^B | 0.238^C | 0.228^A | 0.288^BD | 0.288^BD | 0.263^ABD | 0.404^BD | 0.320^D | 0.684 | 0.311^ABD | 0.477^BD | 0.703 | 0.135 | 0.582 | 0.716^ABD | 0.327^ABD |
| LCE | Ensemble | 0.352 | 0.164 | 0.381^A | 0.349 | 0.453^C | 0.276 | 0.221^ABC | 0.326^A | 0.297^A | 0.260^A | 0.247^A | 0.288^AB | 0.303^BC | 0.272^ABC | 0.414^BC | 0.367^B | 0.694 | 0.322^ABC | 0.474^BC | 0.746 | 0.151 | 0.661^AB | 0.700^ABC | 0.361^ABC |
| RankNet | Random | 0.362^BCD | 0.177 | 0.370^BCD | 0.376^BCD | 0.461^BCD | 0.280^BCD | 0.216^BCD | 0.334 | 0.309^BCD | 0.254^BCD | 0.258 | 0.324^BCD | 0.240 | 0.309 | 0.307 | 0.598 | 0.338^BCD | 0.353 | 0.633 | 0.149 | 0.672^BCD | 0.628 | 0.313 |
| RankNet | BM25 | 0.362^ACD | 0.260 | 0.379^ACD | 0.381^ACD | 0.474^ACD | 0.291^ACD | 0.219^ACD | 0.348^CD | 0.265^ACD | 0.249^CD | 0.304^CD | 0.313^ACD | 0.277^D | 0.422^CD | 0.368^CD | 0.720 | 0.342^CD | 0.482^C | 0.785 | 0.154 | 0.767^CD | 0.707^CD | 0.375^CD |
| RankNet | CE | 0.369^ABD | 0.253^D | 0.381^ABD | 0.376^ABD | 0.473^ABD | 0.288^ABD | 0.219^ABD | 0.351^BD | 0.313^ABD | 0.272^ABD | 0.247^BD | 0.308^ABD | 0.313^ABD | 0.280^BD | 0.410^BD | 0.372^BD | 0.717 | 0.339^ABD | 0.482^B | 0.787 | 0.152^D | 0.701^ABD | 0.665^BD | 0.367^BD |
| RankNet | Ensemble | 0.305 | 0.182 | 0.368^BD | 0.367^BCD | 0.469^ABD | 0.279^BD | 0.217 | 0.284 | 0.333^BCD | 0.303^ABCD | 0.268^BCD | 0.237 | 0.305^BCD | 0.259^BCD | 0.403^BCD | 0.338 | 0.655 | 0.343^BCD | 0.451 | 0.691 | 0.148 | 0.680^BCD | 0.713^BCD | 0.340^BCD |
| mMSE | Random | 0.348 | 0.249 | 0.387^D | 0.363^BCD | 0.464^BCD | 0.263^D | 0.334^D | 0.299^CD | 0.263^D | 0.243 | 0.303^CD | 0.301^ACD | 0.309^CD | 0.259^CD | 0.421^CD | 0.374^CD | 0.707 | 0.342^BCD | 0.485^C | 0.782 | 0.150 | 0.686^CD | 0.726^BCD | 0.360^BCD |
| mMSE | BM25 | 0.357 | 0.240 | 0.390^BD | 0.371^ABD | 0.470^ABD | 0.289^BD | 0.222^BD | 0.334^ABC | 0.309^ABD | 0.261^ABD | 0.247^D | 0.306^BD | 0.311^ABD | 0.275^BD | 0.416^BD | 0.376^BD | 0.706 | 0.346^BD | 0.488^B | 0.788 | 0.154^D | 0.703^ABD | 0.714^BD | 0.356^ABD |
| mMSE | CE | 0.367 | 0.217 | 0.384^ABC | 0.374^ABC | 0.457^ABC | 0.283^ABC | 0.218^ABC | 0.334^ABC | 0.309^ABC | 0.268^ABC | 0.247 | 0.297^BC | 0.305^ABC | 0.275^ABC | 0.421^ABC | 0.376^BC | 0.708 | 0.341^ABC | 0.480 | 0.793 | 0.155^C | 0.699^ABC | 0.678^BCD | 0.379^ABC |
| KL | Random | 0.342 | 0.180 | 0.403^BCD | 0.375 | 0.491 | 0.310 | 0.231^ACD | 0.351 | 0.318 | 0.255 | 0.255 | 0.310^BC | 0.332 | 0.283^BCD | 0.383 | 0.324 | 0.630 | 0.340^BD | 0.422 | 0.822 | 0.146 | 0.664^BCD | 0.734^BCD | 0.341^BCD |
| KL | BM25 | 0.287 | 0.204^C | 0.381^ACD | 0.331 | 0.463^CD | 0.230^CD | 0.214^ACD | 0.327^CD | 0.297^CD | 0.251^ACD | 0.240^CD | 0.299^ACD | 0.313^ABD | 0.271^ACD | 0.411^CD | 0.356^CD | 0.690 | 0.339^ACD | 0.482^C | 0.756 | 0.149^D | 0.682^ACD | 0.734^ACD | 0.341^ACD |
| KL | CE | 0.320 | 0.204^B | 0.387^ABD | 0.345^D | 0.469^BD | 0.278^BD | 0.222^ABD | 0.331^BD | 0.263^ABD | 0.243^BD | 0.307 | 0.304^BD | 0.270^ABC | 0.418^BC | 0.364^AD | 0.698 | 0.338^ABD | 0.481^B | 0.773 | 0.151 | 0.680^ABD | 0.704^ABD | 0.351^ABD |
| KL | Ensemble | 0.333 | 0.187 | 0.382^ABC | 0.347^C | 0.457^BC | 0.282^BC | 0.215^ABC | 0.323^BC | 0.292^BC | 0.257^ABC | 0.241^BC | 0.286 | 0.300^BC | 0.270^ABC | 0.416^BC | 0.367^BC | – | 0.341^ABC | 0.476 | 0.761 | 0.149^B | 0.671^ABC | 0.701^ABC | 0.395^ABC |

## G ADDITIONAL FIGURES

As a qualitative way to discriminate between domains, observe across Figure 4 the increasing alignment of model scores to a power law as harder negatives are applied. Under increasingly difficult negatives, the model's score distribution stretches into a heavy-tailed, near–power-law form: only a few distractors receive high scores, while other documents are driven sharply downward. The slope of this tail offers a simple, domain-agnostic measure of how confidently the model assigns relevance. Importantly, when evaluated out of domain, this power-law alignment vanishes entirely, potentially indicating overfitting, as indicated by reduced effectiveness out of domain in our main findings.

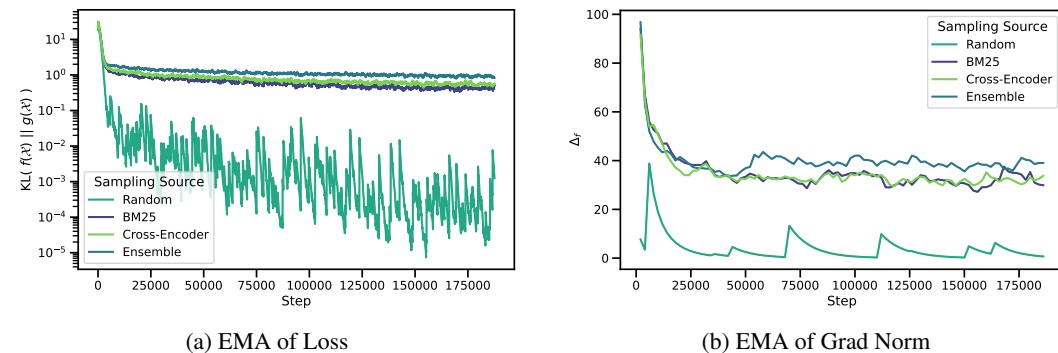

|   (a) EMA of Loss   |   (b) EMA of Grad Norm   |

Figure 3: Training loss in the form of the KL divergence (left) and gradient norm of the student $f$(right). Note the log scale of loss values. Observe the reduced variance in gradient under locality but otherwise minimal difference between sampling domains, we observe that loss converges marginally higher inversely with density ratio $\kappa_Q$.

## H  LICENSES

**Datasets** MSMARCO is licensed under the MIT license, strictly for non-commercial research purposes. NQ and DBPedia are provided under the CC BY-SA 3.0 license. ArguAna and Touché-2020 are provided under the CC BY 4.0 license. CQADupStack is provided under the Apache License 2.0 license. SciFact is provided under the CC BY-NC 2.0 license. SCIDOCS is provided under the GNU General Public License v3.0 license. HotpotQA is provided under the CC BY-SA 4.0 license. TREC-Covid test queries and judgements are provided under open domain however the underlying CORD-19 collection has variable licensing and we point readers to this metadata for more details.

**Models** All base checkpoints (BERT and ELECTRA) are provided under Apache-2.0.

## I  USAGE OF GENERATIVE AI

Within this work, we employed generative AI to critique the manuscript, particularly to improve the narrative in our introduction, through rounds of summarisation to ensure key points were clearly stated. Additionally, a generative agent facilitated through the open-source AI2 Asta was employed to explore literature and ensure comprehensive coverage.

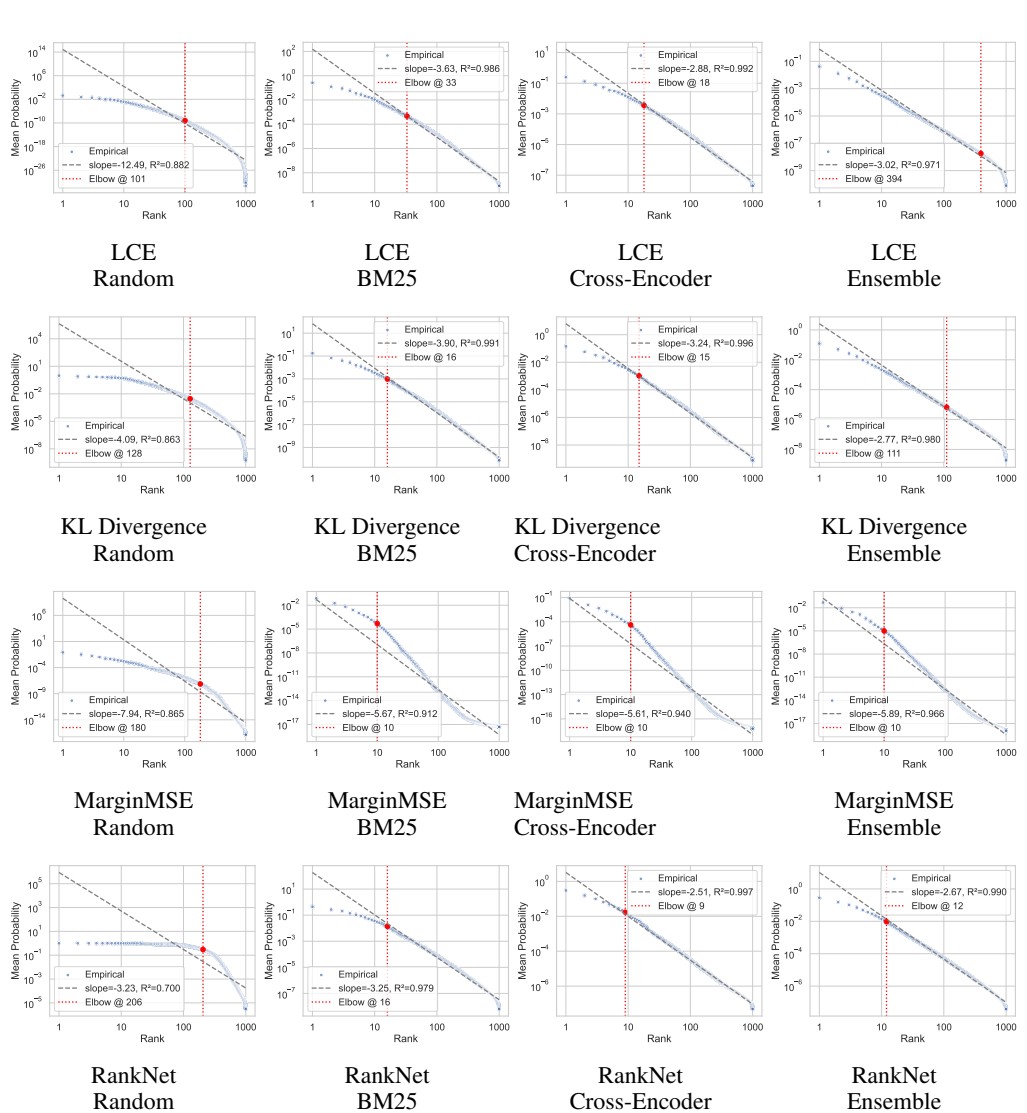

Figure 4: log-log plots of average score at each document rank on MS MARCO passage (TREC DL 2019 judged) for each loss function (rows) and domain (columns) when training a cross-encoder.

