# OpenReview forum: "Disentangling Locality and Entropy in Ranking Distillation"
_ICLR.cc/2026/Conference — Submitted to ICLR 2026_

### Official Review · Reviewer_jvhL · 2025-10-17

**Soundness:** 3
**Presentation:** 3
**Contribution:** 3
**Rating:** 8
**Confidence:** 4

**Summary:**

The paper investigates the complexities of training ranking models, particularly focusing on the interplay between sampling strategies and distillation processes in neural ranking systems. The authors conduct a comprehensive study to elucidate the effects of example selection and teacher model entropy on ranking model optimization. They derive theoretical insights into the geometry of model training and provide empirical evidence.

**Strengths:**

1. The paper presents a generalization bound that connects data selection factors to model effectiveness.
2. The authors conduct extensive experiments across multiple benchmarks (TREC Deep Learning 2019/2020 and BEIR) to validate their claims, demonstrating the robustness of their conclusions.

**Weaknesses:**

1. The connection between the theoretical analysis and the empirical experiments could be better explained. While I gradually understand their connections when reading the experiment analysis, I was almost lost at the end of the theoretical part, not sure where the theory would lead us to.
2. Similar to the previous point, this paper would benefit a lot from a more detailed discussion on the implications and potential applications of the theoretical and empirical discovery.
3. While the authors criticize existing works on multi-stage distillation training for the waste of energy, there isn’t a direct comparison or analysis of the energy cost between different systems and how the discovery of this paper could help. While multiple-stage training is generally not favored, this may not be a huge concern if the models are not large (much smaller than general LLMs) and can converge fast.

**Questions:**

From what perspectives the discovery can directly help us reduce carbon footprint for retrieval model training? Any detailed experiments?

---

> ### Author Response · Authors · 2025-11-21
> **Response to the Reviewer**
>
> We thank the reviewer for their comments and note of our efforts towards robust empirical and theoretical conclusions. We will now clarify the reviewer’s questions.
>
> *Bridging Theory to Empirical Observations*
>
> Our theoretical contribution bridges to empirical observations as it allows for current intuitions which are primarily qualitative, e.g “hard negatives are those that look relevant to a model but are not,” and quantifies them in terms of measurable geometric properties of a ranking. We show that when the associated space of a given query is not tightened, hard negatives fail to further improve over simpler approaches. The theory leads us to focus on quantifiable measures as opposed to purely intuition, and we hope that beyond our own initial exploration of entropy control (see our experiments in entropy ranges), future work can build more principled data selection strategies within ranking.
>
> *Energy/carbon implications*
>
> The measurement of exact energy usage per model not only varies by training setting and compute setting (not public for many negative sources and more generally academic works) but also the region in which compute is executed, thus within a work such as this, exact values for such measures are difficult to generalise, unlike benchmarks and theoretical prescriptions. We agree that such measurements are important and partially underpin this work; however, we would leave this study to those more equipped to measure across multiple regions.
>
> For the reviewers' interest, however, consider the difference between a single low-core CPU node, which can be used to mine BM25 negatives, versus current ensembling methods. To reproduce current filtered ensembles, use the 5th iteration, where each model of 12 (excluding BM25 and TAS-B) has used negatives from previous iterations and distillation data. The difference is large enough to be intuitive, such that when no measurable effectiveness gain can be found, such computational expense is not justified.

---

> > ### Comment · Reviewer_jvhL · 2025-11-26
> >
> > Thanks for your response. I acknolwedge that it's hard to do the energy analysis within one week, but I suggest the authors to add it in the final version of the paper, e.g., given same budgets, what's the performance differences, or given the same performance, what's the budget difference. I will keep my score unchanged.

---

> > > ### Author Response · Authors · 2025-12-03
> > > **Response to the Reviewer**
> > >
> > > We thank the reviewer for taking the time to respond to our clarifications.
> > >
> > > Given the statistical equivalence of multiple settings under distillation, we could provide approximations of carbon usage deltas in a reference setting, which could further motivate our work in a camera-ready copy. We will further consider this point and consult carbon-use literature to ensure that such approximations are generalisable and not misleading. We thank the reviewer again for the suggestion.

---

### Official Review · Reviewer_xeEP · 2025-11-01

**Soundness:** 3
**Presentation:** 3
**Contribution:** 2
**Rating:** 6
**Confidence:** 2

**Summary:**

This paper studies to disentangle two ingredients in neural ranking training – locality (data sampling) and entropy (from teacher).  The authors point out that the modern neural ranking models heavily rely on hard-negative mining with multiple stages without understanding the two key aspects that could independently contribute to the resulting model performance. The paper proposes a theoretical framework to describe how these two factors (locality and entropy) affect generalization in ranking distillation. Through comprehensive ablations, the authors show a) elaborate sampling pipelines yield negligible improvements over simple heuristics under distillation, and (b) moderate teacher entropy improves performance, while overly confident or noisy teachers degrade it.

**Strengths:**

* Intriguing study that questions the standard recipe (multi-stage hard negative mining) for neural ranking and seeks a deeper, more principled understanding of its training dynamics.
* Presents a theoretical framework that rigorously examines generalization bounds through the lenses of locality and entropy.
* Offers evidences that once locality and entropy are properly managed, complex multi-stage training pipelines add limited value.

**Weaknesses:**

* The experimental configuration is somewhat narrow, particularly in its treatment of negative sampling. The paper relies on simplified strategies rather than the more sophisticated dual-encoder mining pipelines commonly used in practice.
* The evaluation is restricted to a limited set of ranking benchmarks, excluding broader or multitask-oriented suites such as MTEB that better capture generalization across domains. As a result, the findings may not fully extend to modern multitask embedding training setups (multi-task, generalization to unseen tasks).
* The second observation regarding entropy has been explored in prior work, as noted below.

**Questions:**

* Did the authors evaluate alternative negative mining strategies such as nearest neighbor retrieval using separate dual encoders, as employed in approaches like RocketQA [1]?
* The second part – overly confident teachers degrade performance was discussed in [2]. Can authors compare their findings with the current findings?

[1] Qu, Yingqi, et al. "RocketQA: An optimized training approach to dense passage retrieval for open-domain question answering." arXiv preprint arXiv:2010.08191 (2020).

[2] Menon, Aditya K., et al. "A statistical perspective on distillation." International Conference on Machine Learning, 2021.

---

> ### Author Response · Authors · 2025-11-21
> **Response to the Reviewer**
>
> We thank the reviewer for their time and acknowledgement of our efforts towards a principled understanding of negative mining. We now clarify the reviewers' questions concerning our contribution.
>
> *Dual-encoder mining (RocketQA-style)*
>
> While we do not include a separate dual-encoder “RocketQA-style” miner in our experiments, such pipelines correspond in our framework to a mining policy nu_Q​ that further tightens Delta_Q and often lowers empirical entropy. Our corollary for biased sampling shows that kappa_Q inflates the rate term while the combined entropy/locality term controls the bias; once Delta_Q​ is sufficiently small, additional stages – whether cross-encoder or dual-encoder based – are predicted to yield diminishing returns unless they substantially improve the entropy profile. We will make this mapping explicit in Section 3.1 and highlight dual-encoder mining as a natural instantiation.
>
> *Relation to Menon et al. (teacher confidence)*
>
> Our empirical result—that over-confident teachers degrade performance while moderate entropy helps—is consistent with statistical perspectives on distillation. We will also clarify in the paper that our contribution is complementary: Menon et al. analyse the calibration of teacher confidence in a classification setting, whereas we focus on pairwise ranking under biased sampling and explicitly couple teacher entropy with locality (Delta_Q​) and sampling bias (kappa_Q​). Our empirical results confirm the “moderate confidence” regime in a setting where both locality and distillation interact.
>
> *Evaluation scope (MTEB)*
>
> Our BEIR suite already spans open-domain QA, argument retrieval, scientific/medical, Quora, NQ, etc. (see Table 5), which substantially overlaps MTEB retrieval tasks. If the reviewer could clarify how residual tasks in MTEB might affect our conclusions, this would be greatly appreciated.

---

### Official Review · Reviewer_xh3B · 2025-11-01

**Soundness:** 3
**Presentation:** 3
**Contribution:** 2
**Rating:** 6
**Confidence:** 2

**Summary:**

This paper investigates the joint influence of negative sampling locality and teacher entropy on neural ranking distillation. The authors derive a generalization bound involving the query manifold diameter and teacher pairwise entropy, providing a theoretical explanation for when hard negatives or complex distillation pipelines are (or are not) beneficial.

**Strengths:**

Hard-negative mining and ranking distillation are widely used; the paper addresses a real gap in IR training methodology.

**Weaknesses:**

1. Theoretical reasoning is largely descriptive, not prescriptive
The generalization bound identifies factors but does not provide actionable guidance for selecting sampling policies or entropy levels in practice.
The theoretical exposition is mathematically sound but overly symbol-heavy.

2. While the study provides clear insight into bi-encoder retrievers trained via distillation, it remains unclear whether the identified effects of locality and teacher entropy extend to LLM-based retrieval settings

**Questions:**

The experimental findings indicate that moderate teacher entropy provides the best performance.
How should practitioners determine or tune this entropy level in practice?

All experiments appear to use a single teacher configuration.
Would the observed conclusions still hold if the teacher were substantially weaker or stronger?

---

> ### Author Response · Authors · 2025-11-21
> **Response to the Review**
>
> We thank the reviewer for their time and for acknowledging that our work fills a real gap in the existing training literature. We will now provide clarification on the reviewer’s questions.
>
> *Theory*
>
> We use theory to properly quantify how training decisions affect model effectiveness, though in this sense, our bounds are descriptive. In providing this description, coupled with our empirical observations, we can see that expensive pipelines can be rendered spurious. Thus, our prescription leads to a more efficient but equally effective training process. Furthermore, in quantifying training factors, our work can lead to more principled developments within ranker training. In particular, the bound suggests that once locality is adequate (e.g., BM25 or a single strong retriever that already tightens Delta_Q​), investing additional compute in deeper negative-mining pipelines has diminishing returns unless it also improves the entropy profile. This directly motivates our ablations: we compare BM25, CE, and an ensemble pipeline under fixed teacher and show that the empirical gains align with the predicted changes in Delta_Q​ and teacher entropy.
>
> *LLM-based Distillation*
>
> Our bound is teacher-agnostic and only requires pairwise preferences to compute H(g); the kappa_Q term simply captures the shift induced by an LLM-based pool. If the reviewer could clarify how our framework would change under LLM distillation, this would be greatly appreciated. If an LLM is used as a teacher, its pairwise preferences simply define a different ggg and thus a different entropy profile \eta(H(g)), while the mining policy nu_Q may also change (e.g., via LLM-augmented candidate pools); our framework can accommodate this by updating kappa_Q​ and H(g) accordingly.
>
> *How should practitioners determine or tune this entropy level in practice?*
>
> In practice, our primary prescription is to reduce the usage of overly complex pipelines for negative mining under a distillation setting. Our experiments already show that even naive controls (pruning instances outside a user-prescribed quartile range) can have a positive effect. Concretely, our investigation provides a theory to guide future work and preliminary evidence validating that entropy values can be tuned with measurable downstream effects.
>
> *Would the observed conclusions still hold if the teacher were substantially weaker or stronger?*
>
> Much like the hard-negative pipelines we ablate, we instead look at quantities that affect training rather than downstream effectiveness or direct intuition. We would instead focus on the entropy profile produced over positives and negatives; an LLM distillation process would change this profile, and the subsequent effect could be estimated (with some looseness) a priori. As mentioned previously, our control over entropy profiles is sufficient to explore this behaviour within our framework; nevertheless, such an empirical investigation is a clear direction for future work.

---

### Official Review · Reviewer_ghpY · 2025-11-02

**Soundness:** 2
**Presentation:** 1
**Contribution:** 2
**Rating:** 4
**Confidence:** 3

**Summary:**

Modern neural ranking systems combine hard negative sampling and knowledge distillation to train models efficiently. Due to sparsity of human annotations, especially in MS MARCO, naive sampling procedures can produce false negatives hurting model’s performance. To overcome this issue, we often rely on expensive methods to filter out false negatives, e.g., SentenceTransformers use 12 cross-encoder models.  Their benefits are poorly understood, as the effects of sampling (data locality) and labeling (teacher entropy) are entangled. The paper aims to disentangle these two influences—locality and entropy—to determine which truly improves ranking performance. To this end, authors:


1. Establish a new generalization bound.
2. Carry out experiments showing that benefits of “smarter” sampling (i.e., with post-filtering) can be limited, which they “connect” to their theoretical result.


Authors also mention that they studied the role of data augmentation, but I found no evidence of this in the paper. Although  I like the direction of work and empirical observation, I found the paper to be extremely confusing, experiments to be somewhat limited, and the bound seems to be vacuous.


I understand that theory is hard and deriving any bounds is a step forward. It is great to have both empirical and theoretical results. However, the narrative “treats” this bound as a tight one, but this is unlikely to be the case. I also suspect that some experimental results are due to training instability rather than genuine trends. I think it can be potentially a great paper, but it requires a thorough revision.

**Strengths:**

1. An important, although well-studied problem.

2. A novel (though potentially vacuous) bound that connects the diameter of the query and the teacher’s entropy to the generalization gap.

3. Paper has both in-domain and out-of-domain experiments.

**Weaknesses:**

1. The paper is quite confusing, it uses inconsistent terminology (while not defying some crucial parts). It also makes quite a few technical claims (although tangential to their main results) that seem to be incorrect. Math is confusing: formulas appear to have errors (e.g., the risk minimization definition) and unexplained symbols (e.g., eta). Data augmentation is mentioned, but never explained properly. More detailed comments are below.

2. The proposed bound  appears to be vacuous because it involves a VC dimension, which is astronomically large in modern neural networks (see references  and discussion in the question section).

3. The experimental results are rather limited. Some of these are nice-to-have experiments, but others are more important:
* Only one training collection is considered. In that MS MARCO is known for its sparse judgements, how about some other collections that are different (e.g. Natural Questions)?. **Important addition**
* Retrieval uses only BM25, but not stronger bi-encoder models: **Crucial addition**
* There is seemingly only one training seed per experimental scenario. It is quite possible that some of the experimental results are due to seed-related randomness: **Crucial to verify**
* Some experimental settings are poorly motivated (see detailed comments).

* Last but not least, authors consider only basic approaches to sample and filter out false negatives. However, quite a few recent approaches were proposed. Some of the use LLM-judges. I don’t suggest trying every new approach, but something stronger than CE-based filtering should be considered IMHO. **This is in the nice-to-have category, i.e., I would not reject the paper if it didn’t have it.**

Some links:
1. Negative Sampling Techniques for Dense Passage Retrieval in a Multilingual Setting
I recommend consulting the following two papers/preprints, which also have references for older approaches as well (ADORE, STAR, AR2, RocketQA):

PROD: Progressive Distillation for Dense Retrieval. Zhenghao Lin, Yeyun Gong, Xiao Liu, Hang Zhang, Chen Lin, Anlei Dong, Jian Jiao, Jingwen Lu, Daxin Jiang, Rangan Majumder, Nan Duan, 2023

TriSampler: A Better Negative Sampling Principle for Dense Retrieval. Zhen Yang, Zhou Shao, Yuxiao Dong, Jie Tang, 2024.



**Detailed comments:**



**Do not respond to this list, unless you feel like it’s a big-time misunderstanding on my side that could change my opinion of the paper. These are just for your information. Questions here are rhetorical. Thank you!**

L069-074. This is not clearly worded. How is epistemic uncertainty relevant here? How does it connect to the appeal of distillation? BTW, it’s not even very clear why it’s an appeal.

> that multiple relevant documents may be present and optimised within a single instance.

It’s not clear what an “instance is”. In the contrastive ranking framework you can surely select multiple positives and multiple negatives per query. I can argue that a loss term (even if it can be additively decomposed into terms with a single positive) is an instance.

L108: is that as we produce an ever stronger source of negatives, due to label sparsity, we inevitably sample false negatives -> stronger source of negatives produces negatives that are strongly negative, doesn’t it?

as such the notion of de-noised negatives has been proposed -> This is again a very vague phrase. You should probably call it “filtering out of false negatives”, de-noising is not sufficiently specific.

Moreover, given that a notion of relevance is, indeed, subjective, the paper would benefit from a short discussion on what “noise” is, how it can negatively affect performance. You should IMHO be more specific that the filtering pipeline is filtering out false negatives rather than some unspecified noise.

See, e.g.:

Nandan Thakur, Crystina Zhang, Xueguang Ma, and Jimmy Lin. 2025. Hard Negatives, Hard Lessons: Revisiting Training Data Quality for Robust Information Retrieval with LLMs. EMNLP Findings.

L117  where a larger number of elements would not be ranked **without being utilised** -> what does it mean?



L131-134 In terms of explicit generalisation bounds, Hsu et al. (2021) provide a bound under uniform convergence, using distillation as a vector for understanding the original teacher model, we diverge from this setting as in downstream Information Retrieval we focus on trading off effectiveness for reduced latency -> I can’t see how these two statements (using distillation as a vector for understanding the original teacher model and a following one) are related. Moreover, it’s not understandable what “distillation as a vector for understanding the original teacher model and a following one” means on its own.

L146-147 I think this is not a grammatical sentence, because it lacks a verb.



L151 I disagree that the ranker is trained as a regression model. Where does this come from? It’s a classification problem with typical classification losses like the margin loss or soft-max-like loss InfoNCE.

L153 I also think it’s incorrect what you claim about bi-encoders. Both bi-encoders and cross-encoders are trained contrastive losses and the problem is still a classification problem. The main difference with bi-encoders is that the relevance score is computed as an inner-product between two vectors, but the loss is still a classification loss.

L157 It’s not clearly what solely positive means.

L158 What is the importance of citing MS MARCO here? The annotation procedures were not introduced by MS MARCO folks.

L163 It’s not clear what the teacher model g() produces: labels or logits?

L165-173 This is hardly understandable. First, why do we call them pseudo-negative? Second, you didn’t mention / describe the sampling approach. You should say that typically negatives are sampled from a top-k returned by a retriever. However, not all of the negatives are true negatives due to the sparsity positive labels (i.e., it is not feasible to find and annotate all positive examples, in particular, because this procedure is retriever-dependent. We can find all positive documents returned by one retriever. However, another retriever may “bring” other top positives to top-k). This is, of course, exacerbated by the problem that you touch upon in the next paragraph: what is a positive and what is a negative is not well defined, i.e., a problem. This will be a good place to describe what you mean by “noise” and not in the next paragraph (problem setting).

Then you should say that there are heuristic approaches to filter out false-negatives (not pseudo-negatives!).

L179 What’s an auxiliary data? In fact, if you use something like a black-box LLM ranker, there is a non-trivial chance it was trained on your test data: TREC DL or BEIR. These are public collections with publicly available test sets.

L186 Metric structure matters fundamentally because -> Which metric are you talking about?

L190-196 This whole paragraph is not really understandable. Without describing RankNet you can’t claim it blurs the boundaries. In fact, I think this not correct, because RankNet is not a distillation loss per se. Virtually any supervised classification loss can be turned into a distillation loss by replacing hard labels with normalized teacher scores (e.g., sigmoids of logits).

In L193 you mention some downstream task and observations. Why do you need a vague term observation when you already defined one as a set of query-document pairs? This is particularly confusing given you don’t describe top-k sampling before and assume that everyone knows it.

L213 Why doesn’t Eq (2) incorporate ground-truth ordering (I think it should!)? Are we actually summing up for all pairs when x is ranked higher than x’ or vice versa?



I have the same question for Eq (4) and Definition A.3 and Definition A.7?



A follow-up question regarding Eq (4): Is it true for any probabilities or only for the RankNet style probabilities defined as a sigmoid of score differences?



L215 Equation (2) seems to be incorrect because it does not incorporate ground-truth ordering.

L216 you need to explain why you deal with generic Bregman divergences and not just KL-divergences. It becomes clear in the experimental section, but that’s too late!

L246 Applying a PAC bound affects model preference for a hypothesis within a class, effectively governing how a model will generalize. -> I hate to nitpick here, but the PAC bound itself doesn’t govern anything.



L320 by optimising the margin between positive (Xi) and negative (Xj ) elements instead of pointwise scores (Hofstätter et al., 2020). -> I think “instead of” is out of place. It sounds like you optimize

L325 “It sees increasing application with increasing precision of modern ranking models as it optimizes all x, x’ interactions agnostic of human labels. -> It is not clear what it means and requires a citation. “Interactions agnostic of human labels” is a particularly dubious definition/claim.





L329 What is f() in Eq 10-12?  Please clarify/fix



What does eta mean in Theorem 2.1? BTW, it’s not bad to remind that g() is a teacher scoring function.



L340 We show these semi-supervised criteria can be expressed as Bregman divergences -> Only now we learn why your theory is applied to Bregman divergences and how distillation is related to the RankNet loss. This should have been defined much earlier!



L342 All models observe a total of 12M documents -> what does it mean to observe?



L350-354 We consider four sampling sources in our investigation, aligning with those applied in literature. The first is uniform selection from the training corpus (Random). The second is a lexical heuristic BM25 (k1 = 1.2, b = 0.75) (Robertson et al., 1995), a lightweight retrieval model. **The third is our teacher model**, a cross-encoder (CE). Finally, we apply the ensemble pipeline shown in Figure 1 (Ensemble).

How is a CE model is a sampling source?


Figure 1: I don’t see an ensemble there.

It is rather confusing than use the term semi-supervision after definining the process as teacher-student learning. I would argue that you should continue to use terms distillation (distillation training) and distillation loss for consistency.

L355 s where investing computational budget in a strong estimator -> Clarify what is an estimator? You probably should use a different term for a false-negative filtering procedure.

L360 You mentioned localized sampling a few times (and also at this point). I believe you never defined it properly.

L369 “empirical values of the query-specific diameter show that the query space does not become more compact.”

Table 1 needs BM25 as a baseline.

I think random is a confusing term, because all negatives are selected randomly. What differs is the selection set and a post-filtering procedure. I would argue that corpus-sampling is a better term than random in Table 1.

**Questions:**

0. Where do you talk about data augmentation? Both empirical and theoretical?

1. There is evidence that VC-dimension of neural networks is astronomical. Doesn’t it make the bound in Eq (8) vacuous? For example, Barlett et al prove it is in the order of WL log(W/L), where W is a number of weights and L is a number of layers. For 10 million neural net with 12 layers, the VC dimension is about 10^10. Even with the number of “observed” data points in the order of one million, the square root in Eq (6) is at least 100. And what do we know about constant C? How large is it?

     * Bartlett, Peter L., et al. "Nearly-tight VC-dimension and pseudo dimension bounds for piecewise linear neural networks." JMLR 2019

     * Zhang et al 2017 (UNDERSTANDING DEEP LEARNING REQUIRES RETHINKING GENERALIZATION)





2. Can you explain the following training irregularities in Table 1? Can they be due to training instabilities, i.e., different seeds would produce very different results?

       * For CE on BEIR and LCE loss corpus-level sampling (denoted as Random) works better? If post-filtering hurts, why does ensemble-sampling work at par with Random?
        * For distillation losses RankNet and KL-div, ensembling actually hurst performance of CE on BEIR. However, this is not the case for MSE. This, again, looks more like a training instability rather than a genuine pattern.


 3. How many seeds did you use per experiment?

4. In Section 2.2 you define a metric-measure space and use an essential supremum. How do you define the measure μQ?

5. Did you train bi-encoders without in-batch negatives? Any comments on this in the paper?

 6. “To ensure reproducibility and clear attribution of effectiveness, we train a cross-encoder following (Pradeep et al., 2022) using the ELECTRA architecture trained for one epoch with BM25 localised negatives on the MSMARCO passage training set.” -> How is this related to reproducibility? What do we try to reproduce? BTW, “BM25-localized negatives” isn’t proper terminology IMHO.

---

> ### Author Response · Authors · 2025-11-21
> **Response to Reviewer**
>
> We thank the reviewer for their thorough review of our manuscript and notes on the importance of our investigation. We now look to address a few core points noted by the reviewer to clarify our contribution. We appreciate the notes of corrections and will incorporate them accordingly. To avoid wasting the reviewers' time, we will address their primary questions.
>
> *Minor Notes*
>
> We primarily look to align our terminology with that of broader literature. The phrasing “locality” has been previously applied in initial works investigating harder negatives [1], and the phrasing “random” aligns with that of broader contrastive literature [2], though we do agree with the reviewer. The Cross-encoder acts as a sampling source by re-ranking a first-stage to depth n and drawing samples from depth k where k << n.
>
> *Data Augmentation*
>
> Throughout the work, we consider the replacement of human labels with model predictions and the replacement of random negatives with those chosen by a heuristic to be forms of data augmentation over training points comprising a query and documents. We will clarify this notion in our introduction.
>
> *Bound scope and “vacuity.”*
>
> In line with the reviewer’s concern, we agree that for large neural networks the capacity term (e.g., via VC-dimension) can be numerically large and therefore not tight. Our aim is not to provide a sharp guarantee for a specific architecture, but to separate the contribution of (i) query-space locality Delta_Q​, (ii) teacher uncertainty, and (iii) a generic capacity term. The qualitative prediction that shrinking Delta_Q and avoiding extreme teacher entropy are helpful, while capacity controls the rate does not depend on the particular numerical value of the capacity constant and is borne out empirically in Tables 1–3. We will also add a short note connecting our findings on teacher confidence to prior work on distillation in classification settings: our findings align with previous observations, but crucially, our analysis is pairwise-ranking specific and couples teacher entropy with locality rather than treating it in isolation.
>
> *3/4. Training Stability and Seeds*
>
> In these experiments, our results and significance are reported for single seeds; this is common in broader ranking literature. With respect to stability, prior art [3] has reported low variance in model effectiveness beyond 1e4 steps in a similar optimisation setting to our own, beyond 1e5 steps; this variance is generally low (particularly in-domain).
>
> *Metric-Measures Spaces for Q*
>
> Concretely, mu_Q​ is the latent query-conditioned distribution over (Q,D) pairs that defines the environment we care about, being relevance (e.g., the distribution induced by user interactions or an idealised pool). Our sampling policy nu_Q​ is a biased approximation of this distribution, and the density-ratio term kappa_Q in our bound precisely quantifies this mismatch. We will make this correspondence explicit in the main body.
>
> *In-Batch Negatives*
>
> To ensure that both cross-encoders and bi-encoders are trained in an identical setting we do not include in-batch negatives; furthermore, under distillation, in-batch negatives, which are low signal already, are not “free” as they are in contrastive settings. Our work aims to provide a controlled comparison across different training settings rather than maximising effectiveness through orthogonal methods.
>
> *Arguing for Reproducibility*
>
> We argue this point primarily as existing sources of hard negatives are often difficult or impossible to reproduce. In the case of an ensemble of negative sources, not only are the particular versions of models often poorly noted, but the public-facing sources are often repeatedly trained upon previous iterations. As noted by Wang et al. [4], reproducing a two-stage negative pipeline can be difficult; thus, we argue that using the filtering of a model that may implicitly apply 5 stages would confound results. Additionally, approaches that apply LLM-as-a-Judge often use closed-source models, thus we cannot fully reproduce these findings. Thus, we train our own cross-encoder using a common setting to prevent such issues.
>
> Citations:
> [1]: Luyu Gao, Zhuyun Dai, and Jamie Callan. Rethink Training of BERT Rerankers in Multi-stage Retrieval Pipeline. ECIR 2021
> [2]: Vladimir Karpukhin, Barlas Oguz, Sewon Min, Patrick Lewis, Ledell Wu, Sergey Edunov, Danqi Chen, and Wen-tau Yih. Dense Passage Retrieval for Open-Domain Question Answering. EMNLP 2020
> [3]: Sophia Althammer, Guido Zuccon, Sebastian Hofstätter, Suzan Verberne, and Allan Hanbury. Annotating Data for Fine-Tuning a Neural Ranker? Current Active Learning Strategies are not Better than Random Selection. SIGIR AP 2023
> [4]: Xiao Wang, Craig Macdonald, Nicola Tonellotto, and Iadh Ounis. Reproducibility, Replicability, and Insights into Dense Multi-Representation Retrieval Models: from ColBERT to Col*. SIGIR 2023

---

> ### Comment · Reviewer_ghpY · 2025-11-26
> **response**
>
> Dear authors thank you for the response.
>
> >We primarily look to align our terminology with that of broader literature. The phrasing “locality” has been previously applied in initial works investigating harder negatives [1], and the phrasing “random” aligns with that of broader contrastive literature [2], though we do agree with the reviewer. The Cross-encoder acts as a sampling source by re-ranking a first-stage to depth n and drawing samples from depth k where k << n.
>
> I have never seen the term locality (although I did read quite a few papers on the topic), so I don't think it is super widely used. It doesn't hurt to define terminology, anyways. The same applies to the "data augmentation" term, especially, given that it is not super-standard to call distillation a data augmentation technique. It is kind of a data-augmentation technique, but **primarily** it is a distillation approach. I would be ok with "data-augmentation through distillation". Note that I did search through the paper for some more traditional augmentation techniques and didn't found, I am not nitpicking about this being confusing.
>
> Again, I am not against this as long as it is properly defined. Random negatives is clearly a misnomer (despite being a somewhat common one), but it is fine to use it as long as you define it. That said, once you define **BOTH** hard negatives and  random negatives as randomly sampled (and your write it down in the paper), the flaw of the "random negative" term becomes obvious to both the reader and the author.
>
> > Bound scope and “vacuity.”
>
> > In line with the reviewer’s concern, we agree that for large neural networks the capacity term (e.g., via VC-dimension) can be numerically large and therefore not tight. Our aim is not to provide a sharp guarantee for a specific architecture, but to separate the contribution of (i) query-space locality Delta_Q​, (ii) teacher uncertainty, and (iii) a generic capacity term. ...
>
> Yes, and if you have a potentially galactic constant, doing this is problematic, IMHO.
>
> > 3/4. Training Stability and Seeds
>
> > In these experiments, our results and significance are reported for single seeds; this is common in broader ranking literature. With respect to stability, prior art [3] has reported low variance in model effectiveness beyond 1e4 steps in a similar optimisation setting to our own, beyond 1e5 steps; this variance is generally low (particularly in-domain).
>
> Sorry about some unclarity. I meant that there's sometimes a substantial variability depending on the seed, e.g, plus-minus 5-10% relative (more like 3-5% in my experience, but it can be more sometimes). The broader ranking literature could have been more thorough. What's particularly concerning in your case: Some results don't make sense intuitively (mentioned in the main review) and this is why I suspect that seeds can be the issue.
>
> > In-Batch Negatives
>
> > To ensure that both cross-encoders and bi-encoders are trained in an identical setting we do not include in-batch negatives; furthermore, under distillation, in-batch negatives, which are low signal already, are not “free” as they are in contrastive settings. Our work aims to provide a controlled comparison across different training settings rather than maximising effectiveness through orthogonal methods.
>
> I generally agree, but I am not 100% convinced using in-batch negatives wouldn't have changed any conclusion.
>
> >Arguing for Reproducibility
> >We argue this point primarily as existing sources of hard negatives are often difficult or impossible to reproduce. In the case of an ensemble of negative sources, not only are the particular versions of models often poorly noted, but the public-facing sources are often repeatedly trained upon previous iterations. As noted by Wang et al. [4], reproducing a two-stage negative pipeline can be difficult; thus, we argue that using the filtering of a model that may implicitly apply 5 stages would confound results. Additionally, approaches that apply LLM-as-a-Judge often use closed-source models, thus we cannot fully reproduce these findings. Thus, we train our own cross-encoder using a common setting to prevent such issues.
>
> This warrants a better clarification, at the very least. If you think that two-stage pipeline is problematic in terms of reproducibility, I think it's sort of a concern. However, there's also a workaround to use an embedding model (or better more than one) trained by other people.
>
> Filtering out negatives with LLM judges are reproducible to a certain degree: you can just release predictions of the LLM judges that you used. Note that you don't have to reproduce somebody else's experiments: You can run a controlled experiment on your own (and share all results).
>
> In summary, I think none of the **individual** concerns are true deal breakers (except perhaps a potential variability across seeds), but together they warrant a revision.

---

> > ### Author Response · Authors · 2025-12-03
> > **Response to the Reviewer**
> >
> > We thank the reviewer for their further discussion points and consideration of our clarifications.
> >
> > Regarding the notion of data augmentation, we appreciate this point and will add the caveat "augmentation through distillation," and agree with the reviewer that a small clarification here can aid broader understanding.
> >
> > The misnomer of random negatives can be rectified via the clarification of uniform sampling from a corpus versus uniform sampling from a biased top-k derived from some estimator or ensemble of estimators. Again, this small clarification is helpful, given that our work aims to better define these traditionally intuition-driven concepts in IR that give rise to misnomers.
> >
> > While we agree that PAC bounds via VC dimension can lead to vacuous constants, one can further tighten these bounds, for instance via Rademacher complexity, though this may provide a tighter bound. In the case of overparameterized language models, such results would likely remain descriptive rather than tight. Thus, under a tighter estimate of this constant, similar conclusions would be drawn: the factors we care about would be defined in the same manner, and our empirical results would not change, nor would our treatment of empirical estimates of factors within our theoretical framework. We appreciate the reviewer's concern and will make this note, as we feel it is important in ensuring clarity within our work.
> >
> > While we understand that effectiveness may be reduced under a given data distribution, we explore a broad set of settings to identify common applications. Our evidence, as noted by the reviewer, suggests that the additional expense in drawing samples from a biased distribution is often poorly allocated. We feel the settings mentioned do not change our conclusions even in isolation, but thank the reviewer for carefully considering the robustness of our results.
> >
> > "workaround to use an embedding model (or better more than one) trained by other people"
> > We politely disagree, as this allows minimal data provenance given the modern landscape of model training, akin to how we have attempted to avoid closed-source models where possible in this work. We have already agreed to use the ensemble of models supplied by the sentence-transformers project, which is known to yield greater effectiveness under contrastive learning but is often employed in distillation, as noted in our work.
> >
> > While the use of LLM judges is indeed a strong ranking source, ultimately, this is similar to using an LLM re-ranker or exhaustive retriever as a negative miner. In contrast, these sources would be stronger; they would require tuning of filtering thresholds using a method such as the one published by Thakur et al. This work was published after our manuscript was submitted. It required a concentrated contribution, which would likely detract from our primary contributions. Furthermore, as a pragmatic point, the tens of thousands spent on LLM judgment experiments are beyond the scope of many academic labs, yet they still focus solely on contrastive learning. Our conclusions suggest that you would better utilise your judgements as labels Y rather than filtering X.
> >
> > We appreciate the reviewers' note that they do not see a deal-breaker in our work and, more broadly, in their reviewing efforts, which will improve our camera-ready version.

---

### Meta-Review · Area_Chair_B6Tk · 2026-01-13

**Summary:**

This paper attempts to disentangle the effects of sampling (data locality) and labeling (teacher entropy) in ranking distillation setting. The authors propose a generalization bound and conduct ablations on MS MARCO to argue that complex, multi-stage negative mining pipelines are often yield minor gains over good sampling strategies under distillation.

This paper had scores across the spectrum. Reviewer ghpY has, in particular, provided a very comprehensive review on the presentation, practicality and empirical analysis. The use of the theoretical analysis in this paper is somewhat arguable. I agree with Reviewer ghpY that the presentation of the paper can be improved significantly. The paper is often dense and theoretical results are not well presented. The empirical analysis is rather limited. I agree with Reviewer ghpY that this paper can be a good paper after a significant revision but it is not ready in the current form. I recommend rejection.

**Reviewer Concerns:**

Reviewer ghpY has raised a number of very important and valid concerns about the paper, that would warrant a significant revision of the paper. The reviewer highlighted that the presentation is very poor. Given that the theoretical contributions form the core component of the paper, this needs to be carefully addressed. The reviewer also highlighted that the empirical analysis is rather limited. The reviewer also provided a number of good points and I believe it is important to address them before submission.

Reviewer xh3B also raised concern about the theoretical analysis of the paper.

**Reviewer Scores:**

Reviewer ghpY would probably decrease the score given their primary concerns are not addressed satisfactorily.

Other reviewers will probably keep their scores. Overall, the paper will be borderline but I think the concerns of Reviewer ghpY need to be addressed carefully.

---

### Decision · Program_Chairs · 2026-01-26

Reject